

# Improving Precipitation Interpolation Using Anisotropic Variograms Derived from Convection-Permitting Regional Climate Model Simulations

Valentin Dura[1,2], Guillaume Evin[2], Anne-Catherine Favre[2], and David Penot[1]

[1]EDF-DTG, Grenoble, France
[2]Univ. Grenoble Alpes, CNRS, INRAE, IRD, Grenoble INP*, IGE, 38000 Grenoble, France
*Institute of Engineering and Management Univ. Grenoble Alpes

**Correspondence:** Valentin Dura (valentin.dura@edf.fr)

**Abstract.**

The consideration of the spatial variability of daily precipitation, assessed through spatial covariance, is crucial for hydrological modeling. Estimating this covariance is particularly challenging in regions with sparse rain gauge networks or limited radar coverage. To address this issue, this study explores the potential of Convection-Permitting Regional Climate Model (CP-RCM) simulations to estimate anisotropic variograms. We compare five approaches: (1) SPAZM, an interpolator based on local precipitation-altitude regressions, Trans-Gaussian Random Fields, differing by their covariance structure and data source with (2) isotropic covariance from rain gauges, (3) anisotropic covariance from rain gauges, (4) isotropic covariance from CP-RCM simulations, and (5) anisotropic covariance from CP-RCM simulations. The models are evaluated with cross-validation and spatial metrics using radar-derived analyses. Results demonstrate that Trans-Gaussian Random Fields outperform SPAZM. Anisotropic covariance models derived from CP-RCM simulations capture orography-induced directional precipitation structures more effectively than the other models, leading to improved interpolation accuracy and better representation of spatial variability. The generated ensemble of conditional simulations successfully reproduces intense precipitation events at the catchment scale, providing valuable uncertainty quantification. For a 17 km$^2$ catchment, mean catchment precipitation can range from 175 mm to 450 mm for a convective event, despite high rain gauge density. These findings highlight the benefits of using CP-RCM simulations to generate anisotropic variograms for probabilistic precipitation interpolation. This approach improves the spatial variability of precipitation, making it highly relevant for hydrological applications such as flood forecasting. Future work will explore the integration of these ensembles into probabilistic hydrological modeling.

## 1 Introduction

Gridded daily precipitation data is essential for various environmental applications, such as assessing flood and drought risks or modeling glacier mass balance. However, rain gauge stations provide sparse and irregular observations in space, necessitating spatial interpolation models to estimate precipitation fields. Common interpolation procedures include local regression (Daly et al., 1994; Gottardi, 2009; Verdin et al., 2016), data assimilation (Alpuim and Barbosa, 1999; Devers et al., 2021; Vernay et al.,



2024), geostatistics (Goovaerts, 2000; Sideris et al., 2014; Guédé et al., 2024), and more recently machine learning (Hengl et al., 2018; Sekulić et al., 2020; Zandi et al., 2022) models. Geostatistical models frequently outperform other statistical models for

precipitation interpolation (Haberlandt, 2007; Bostan et al., 2012; Masson and Frei, 2014). They are considered Best Linear Unbiased Estimator (BLUE) methods (Rao et al., 1973), minimizing error variance but often producing excessive spatial smoothing that underestimates high precipitation intensities (Hofstra et al., 2010; Hiebl and Frei, 2018). To mitigate this issue, researchers use ensembles of equiprobable fields, known as conditional simulations (Frei and Isotta, 2019; Yan et al., 2021), all consistent with the measurements and the observational variance while displaying distinct spatial patterns.

Daily precipitation exhibits spatial autocorrelation as neighboring rain gauges often record similar values (Tobler, 1970). Spatial autocorrelation is mathematically modeled through the variogram (Cressie, 1991), which links spatial distance to observational variability. Traditionally, variograms are assumed to be isotropic, ignoring directional variations due to the limited number of rain gauge stations or for modeling simplicity (Adhikary et al., 2017). However, in complex topographic regions, anisotropy can arise from interactions between atmospheric conditions and mountain ranges (Tobin et al., 2011), degrad-

ing interpolation quality with isotropic variograms. To improve variogram estimation, researchers have explored alternative sources of spatial information, such as gridded precipitation products. Radar data has been used for deriving parametric or non-parametric variograms (Velasco-Forero et al., 2009; Schiemann et al., 2011) but is often available for shorter timeframes than the multidecadal period required for the daily precipitation analyses. Alternatively, simulations from Convection-Permitting Regional Climate Models (CP-RCMs) (Rockel et al., 2008; Brousseau et al., 2016; Keuler et al., 2016; Gerber et al., 2018) are

supplied on extended periods and deliver new possibilities to infer variogram estimation. Although daily CP-RCM precipitation fields are frequently biased (Caillaud et al., 2021), they might still provide relevant information about anisotropy structure. This study evaluates the accuracy of daily precipitation interpolation by comparing isotropic and anisotropic variograms derived from rain gauges. It also investigates the potential of CP-RCM simulations for deriving anisotropic variograms, a novel approach that has not been previously explored. Additionally, the study aims to quantify the uncertainty in spatial interpolation

at the catchment scale, a critical factor for hydrological modeling, which is rarely found in precipitation analyses or reanalysis (Frei and Isotta, 2019; Devers et al., 2021, 2024).

To address these objectives, we evaluate the ensemble means (kriging) and ensemble spreads (conditional simulations) from geostatistical models in a cross-validation framework across 786 intense precipitation events. We compare four types of variogram: (1) isotropic variogram estimated with rain gauge stations, (2) anisotropic variogram estimated with rain gauge

stations, (3) isotropic variogram estimated with CP-RCM simulations, and (4) anisotropic variogram estimated with CP-RCM simulations. The geostatistical models are also compared to SPAZM (Gottardi, 2009), an interpolator based on local precipitation-altitude regressions stratified by weather patterns. Beyond point-scaled validation, we assess spatial structure using radar-derived analyses as a reference and evaluate the ability of the ensemble spread to capture mean catchment precipitation, an essential factor for hydrological modeling. The analysis is conducted over a mountainous region near the French

Mediterranean Sea. This area encounters heavy rainfall due to the interaction of the air masses with elevation, offering an ideal setting for the study. The paper is structured as follows: Section 2 describes the domain under study and the available precipitation datasets. Section 3 introduces the four covariance modelings used for spatial interpolation. Section 4 displays the





cross-validation and spatial evaluation results. Section 5 compares the findings to the existing literature and discusses future perspectives. Finally, Section 6 outlines the conclusions.

## 2 Domain and data under study

### 2.1 Study domain

The present study focuses on the Cevennes region in the southern Central Massif (see Fig.1). This mountainous study area is characterized by rugged terrain, including high plateaus and forested hills, with elevations reaching 1,700 m. Annual precipitation experiences significant spatial variability. The higher elevations can collect more than 2,000 mm while the lower foothills present a Mediterranean climate with annual precipitations ranging from 600 to 1,200 mm (Canellas et al., 2014). Winter precipitation can fall as snow at high elevations, even if snow cover is intermittent. Autumn is the wettest season. The moisture-laden air from the warm Mediterranean Sea faces colder air from the north, leading to frequent extreme rainfall events of 300-500 mm and generating heavy floods in the valleys (Delrieu et al., 2005).

To capture the spatial variability of precipitation induced by the complex topography, a dense network of 973 rain gauges is available, including 197 stations in the study domain and 776 additional stations near the borders of the study domain. Rain gauges are up to 1,500 m, encompassing the full elevation distribution. The number of available rain gauges on a daily timescale varies over the years, ranging from a minimum of 625 to a maximum of 836.

| ID | Name | Surface (km$^2$) | Mean altitude (m) | Maximum altitude (m) |
|----|------|------------------|-------------------|----------------------|
| A | Altier at la Goulette | 106 | 1156 | 1629 |
| B | Chassezac at Pont Du Mas | 51 | 1246 | 1440 |
| C | Chassezac at Sainte-Marguerite | 413 | 1056 | 1629 |
| D | Thines at Pont de Gournier | 17 | 878 | 1065 |

**Table 1.** ID, name, surface area in km$^2$, mean and maximum altitude in m of the four catchments. Figure 1 shows the locations of the catchments.

### 2.2 Meteorological data

#### 2.2.1 COMEPHORE

The product called COmbinaison en vue de la Meilleure Estimation de la Precipitation HOraiRE (COMEPHORE) (Champeaux et al., 2009) is a high-resolution precipitation analysis product for France, developed at Meteo-France, and based on radar-gauge merging to generate spatial precipitation estimates at an hourly timescale with a spatial resolution of 1 km. COMEPHORE is assumed as a reference to assess the spatial variability of precipitation, taking advantage of the radar coverage. COMEPHORE covers the 1997 - 2025 period but presents a methodology change in 2007, which separates interpolation for convective and stratiform precipitation. It uses the radar corrected by rain gauges to estimate convective precipitation and performs ordinary kriging of the rain gauge amounts for stratiform precipitation using an isotropic variogram. Using radar data,





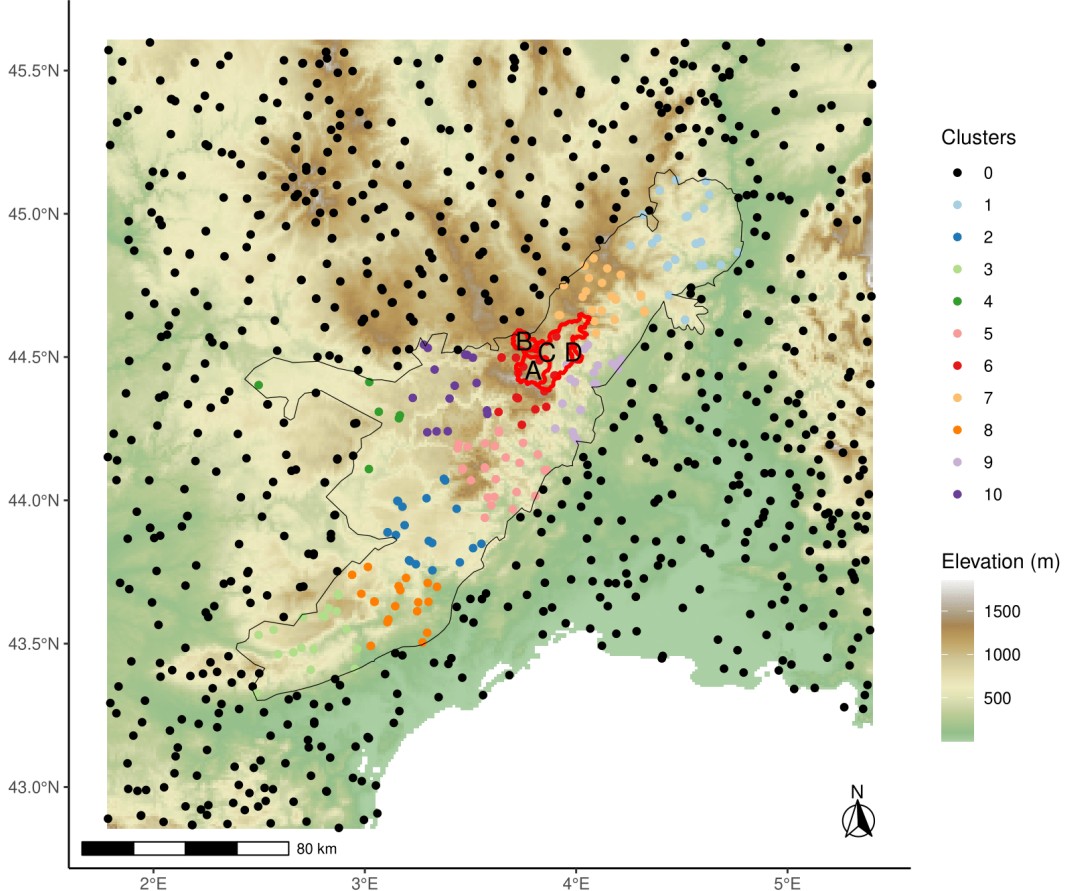

**Figure 1.** Study domain (1.7—5.5°E, 42.8–45.6°N), colored by the elevation (height above sea level in meters), borders of the Cevennes region (black lines), borders and ID of four catchments (red lines, used in sections 3–5), and location of rain gauge stations (dots, around 800 stations available per day). The colors of the stations (dots) are explained in Section 3. Table 1 gathers information about the catchments.

COMEPHORE can accurately represent the size and direction of precipitation cells, and the very local storm cells that the rain gauge network could miss. For this study, we aggregated the COMEPHORE hourly precipitation fields at the daily timescale over the 2008 - 2017 period. Some studies use COMEPHORE as a reference to evaluate climate models (Caillaud et al., 2021), and others use it for hydrological modeling (Evin et al., 2024).

### 2.2.2 AROME

AROME simulations (Caillaud et al., 2021) are generated through the CP-RCM AROME, using hourly atmospheric inputs provided by a CNRM-ALADIN RCM simulation (Nabat et al., 2020), which is driven by ERA-Interim reanalysis (Dee et al., 2011). The AROME simulations are available at the hourly timescale for the Alpine region, as described in the Flagship Pilot

Study of the Coordinated Regional Climate Downscaling Experiment (CORDEX-FPS,Fantini et al. (2018)), at 2.5 km spatial resolution, and cover the 1982–2018 years. In the study, AROME simulations are aggregated to a daily timescale. Unlike RCM models, the AROME model fully resolves convection processes, removing parametrization. The 2.5 km resolution allows for capturing fine-scale atmospheric processes. Previous studies (Ban et al., 2021; Caillaud et al., 2021; Monteiro et al., 2022) have demonstrated that AROME provides an improved representation of intense precipitation events compared to the RCM-ALADIN model despite biases in the size and location of convective cells.

### 2.2.3 SPAZM

SPAZM (Gottardi, 2009) creates daily precipitation analyses at 1 km spatial resolution over a large part of France since 1948 and is continually updated by Electricité de France (EDF) for hydrological modeling. The daily precipitation fields constitute an adjustment of a climatological background field using the daily rain gauge amounts. A climatological background field represents the average daily structure of the precipitation field, where the influence of topography is easier to account for. The background fields incorporate local orographic effects conditioned by eight weather patterns. SPAZM leads to balanced water budgets at the annual scale, even in mountainous areas (Ruelland, 2020). However, the spatial variability of the daily fields is questionable, with an unrealistic correlation to the altitude. Moreover, SPAZM underestimates the spatial variance of intense precipitation, resulting in overly smoothed precipitation fields (Penot, 2014).

## 3 Methods

### 3.1 Spatial Interpolation

This study develops a probabilistic geostatistical framework for daily precipitation interpolation, using four variograms derived from rain gauges and CP-RCM simulations. Unlike traditional approaches that rely solely on rain gauge data, our method exploits CP-RCM fields to estimate daily isotropic and anisotropic covariance structures. In the following, we detail the stochastic modeling and evaluation procedures.

### 3.1.1 Data transformation

Geostatistics involving kriging methods are built in the Gaussian paradigm (Diggle et al., 2003). However, daily precipitation data often exhibits a positively skewed distribution due to left truncation at zero precipitation and the presence of high values. To address this issue, we assume the positive precipitation follows a Gamma distribution and apply quantile-quantile transformation (Gyasi-Agyei, 2018) to normalize the data each day. We perform Gamma fitting using the maximum-likelihood approach from the *fitdistr* ℝ package (Delignette-Muller and Dutang, 2015). Spatial interpolation is then conducted in the Gaussian domain. Skewness and heteroscedasticity of the precipitation data are thus explicitly considered (Erdin et al., 2012), theoretically resulting in greater interpolation uncertainty in areas with high precipitation.

The modelization steps include:





1. Gamma to Normal transformation of positive precipitation:

   $Y^* = \Phi^{-1}[F_\gamma(Y)]$, where Y represents daily positive observed precipitations, $F_\gamma$ the distribution function of a Gamma distribution, $\Phi^{-1}$ is the quantile function of a Gaussian distribution with zero mean and unit variance, and $Y^*$ are the normalized precipitations.

2. Gamma to Normal transformation of zero precipitation:

   The zeros are transformed to $\Phi^{-1}[F_\gamma(u_0)]$ with $u_0$ the probability of zero precipitation, estimated by the number of rain gauges with less than 0.5 mm over the total number of rain gauges.

3. Conditional simulations from geostatistical models (detailed in subsection 3.1.2).

4. Back-Transformation of precipitation predictions into Gamma distribution:

   $\hat{Y} = F_\gamma^{-1}[\Phi(\hat{Z})]$. After the back-transformation step, all values lower than $F_\gamma^{-1}(u_0)$ are set to 0.

Once the precipitation data is normalized, we model its spatial structure using geostatistical models, focusing on estimating covariance structures through variogram fitting.

### 3.1.2 Geostatistic models

Precipitation should be centered in the first instance to estimate its covariance structure accurately (Schabenberger and Gotway, 2005). The expectation of the Trans-Gaussian Random Field is modeled in the Gaussian domain as a function of geographical predictors (longitude, latitude, and altitude) and a seasonal climatological background field. The literature commonly employs geographical predictors such as altitude in kriging with external drift model (Bárdossy and Pegram, 2013). The seasonal climatological background fields have been built through Random Forest modelization using CP-RCM simulations in (Dura et al., 2024). Their construction is based on the idea that the fine-resolution simulations of climate models should summarize the topographical influence on precipitation. Climatological background fields have been used broadly (see e.g. Hunter and Meentemeyer, 2005; Gottardi, 2009; Masson and Frei, 2014) to summarize long-term precipitation patterns influenced by orography, improving the robustness of daily precipitation interpolation. We successfully check the absence of multicollinearity by computing the correlation matrix among the predictors.

Spatial interpolation requires the estimation of a spatial covariance. We fit each day four different exponential variograms, all including a nugget effect:

- an isotropic variogram estimated with rain gauge observations (rgISO),

- an anisotropic variogram estimated with rain gauge observations (rgANISO),

- an isotropic variogram derived from daily AROME precipitation field (arISO),

- an anisotropic variogram derived from daily AROME precipitation field (arANISO).




The nugget parameter corresponds to micro-scale precipitation variability and measurement errors. The exponential choice is standard in numerous geostatistical studies (Masson and Frei, 2014; Frei and Isotta, 2019) because it effectively captures the gradual decline in autocorrelation as the separation distance increases. The weighted least-square estimation of the variograms is done in the Gaussian domain for both rain gauge and AROME precipitations. The weights are proportional to the number of pairs observations and inversely proportional to the squared average distance between them. The estimation process only uses 25 % (randomly selected) of the AROME grid cells for computational reasons. We note $\eta$ and $\theta$ as the estimated anisotropy ratio and angle. Schiemann et al. (2011) describes the methodology in more detail, with an ordinary least square estimation in contrast to a weighted least square estimation. The variograms are estimated with the *gstat* ® packages (Pebesma, 2004).

### 3.1.3 Conditional simulations

Conditional simulation of Trans-Gaussian Random Field is conducted using sequential Gaussian simulations implemented in the *gstat* ® package (Pebesma, 2004) to create each day an ensemble of 100 simulations. Gyasi-Agyei (2018) states that the conditional simulations obtained with *gstat* are too granular. They decided to apply a $3 \times 3$ window smoothing technique. We do not apply the same post-processing step because we will, in future developments, employ mean catchment precipitation, which is not sensitive to the granularity. While this ensemble size is sufficient for representing precipitation variability (Frei and Isotta, 2019), it does not account for all sources of uncertainty, such as covariance parameters inference or precipitation undercatch (see subsection 5.3).

### 3.2 Evaluation

In our analysis, the ensemble mean of the conditional simulations is equivalent to the result obtained using deterministic kriging with external drift (Schabenberger and Gotway, 2005) (KED). Later in the article, we will use the terms "kriging" and "ensemble mean" interchangeably. While kriging captures the most likely precipitation value, the conditional simulations capture the entire ensemble of possible realizations, providing spatial uncertainty. To assess the performance of the spatial interpolation models, we evaluate both deterministic and probabilistic interpolated precipitation fields. The evaluation focuses on three key aspects:

- point-scale accuracy using cross-validation on rain gauge stations ;

- spatial structure by comparing interpolated fields with radar-derived precipitation analyses ;

- uncertainty quantification by analyzing the ensemble spread of conditional simulations at the catchment scale.

The models are evaluated on 786 precipitation events (nearly 20 events per year), defined as the days with at least 50 mm recorded at a minimum of five rain gauges.





### 3.2.1 Cross-validation of ensemble means

The deterministic performance of the spatial interpolation models is evaluated through a leave-cluster-out cross-validation scheme. Unlike traditional leave-one-out cross-validation, this procedure removes entire clusters of neighboring stations to better mimic ungauged conditions and accentuate performance differences between the models. An accurate covariance esti-
mation is crucial for sparse rain gauge station networks. Sequentially, a cluster of neighboring stations is removed from the training dataset and then predicted. Ten groups of stations are created by the K-means algorithm (MacQueen, 1967). The clustering variables are the longitude and the latitude of the stations. Figure 1 indicates the rain gauge station clusters.

The evaluated models include (1) SPAZM as the deterministic interpolation model, (2) rgISO: Kriging with an external drift with an isotropic variogram estimated from rain gauge data, (3) rgANISO: Kriging with an external drift with an anisotropic
variogram estimated from rain gauge data, (4) arISO: Kriging with an external drift with an isotropic variogram derived from CP-RCM precipitation fields, (5) arANISO: Kriging with an external drift with an anisotropic variogram derived from CP-RCM precipitation fields. Mean error (ME) measures bias, indicating systematic over- or underestimation. Mean absolute error (MAE) quantifies overall prediction accuracy.

### 3.2.2 Spatial Structure Evaluation with Radar Analyses

Beyond point-scale accuracy, we assess whether interpolated precipitation fields reproduce the true observed spatial variability. The Teweles–Wobus Score (TWS, Teweles and Wobus (1954)) is used to compare the spatial gradients of interpolated precipitation fields with those from the radar-based COMEPHORE fields. Subsection 2.2.1 describes the COMEPHORE data. We consider COMEPHORE fields as a reference in the study domain because of the good radar coverage, allowing an ac-
curate representation of the size and direction of the heavy precipitation cells. It should be noted that for stratiform events, COMEPHORE could favor geostatistic models with isotropic covariance by construction.

TWS is defined as:

$$\text{TWS} = \frac{\sum_{(s,s')\in\text{Adj}} |(p_s - p_{s'}) - (\hat{p}_s - \hat{p}_{s'})|}{2\sum_{(s,s')\in\text{Adj}} \max\left(|p_s - p_{s'}|, |\hat{p}_s - \hat{p}_{s'}|\right)},$$

where $\text{Adj}$ is the set of adjacent grid points $(s, s')$ in the northern-southern and eastern-western direction within the study domain, $p_s$ ($\hat{p}_s$) is the COMEPHORE (predicted) daily precipitation amount at the grid point $s$ divided by the maximum grid amount. In this part, the interpolated fields are obtained using all rain gauges in the study domain. This metric is particularly
relevant for evaluating the performance of variograms in capturing directional precipitation patterns. Lower values of TWS indicate better agreement with radar-based spatial structures.

### 3.2.3 Probabilistic Performances and Uncertainty Quantification

The ensemble spread of conditional simulations provides uncertainty estimates for interpolated precipitation fields. We evaluate whether these uncertainty estimates are both statistically reliable and hydrologically relevant.

We evaluate probabilistic performances from the geostatistical models (rgISO, rgANISO, arISO, arANISO) using Continuous Ranked Probability Score (Matheson and Winkler, 1976) (CRPS) metrics on the 786 precipitation events using the





| Metric | Evaluates | Application |
|--------|-----------|-------------|
| ME | Bias in predictions | Kriging (ensemble mean) + SPAZM |
| MAE | Overall accuracy | Kriging (ensemble mean) + SPAZM |
| TWS | Spatial variability | Kriging (ensemble mean) + SPAZM |
| CRPS | Probabilistic reliability | Full ensemble (conditional simulations) |

**Table 2.** Metrics computed for evaluation.

same leave-one-cluster out procedure. A lower CRPS indicates better probabilistic distribution. To assess whether interpolation uncertainty translates into hydrologically realistic precipitation estimates, we compare, for the 20 most intense events, the ensemble spread of mean catchment precipitation (from conditional simulations) to the radar-derived mean catchment precipitation (from COMEPHORE). A reliable spatial interpolation model should ensure that the observed catchment precipitation falls within the simulated uncertainty range.

Table 2 summarizes the considered metrics.

### 3.2.4 Case studies

We illustrate applications of the spatial interpolation models to provide concrete examples of how variograms affect precipitation interpolation. We show ensemble means, a sample of conditional simulations, and fitted anisotropic variograms for the 2008-05-26 and the 2014-09-18 days. The first event is an example of a South-North anisotropic event. The second is a Cevenol episode, with stationary storm cells, resulting in 300–400 mm precipitation amount in the first foothills of the Cevennes region.

## 4 Results

We conduct cross-validation at rain gauge stations and evaluate spatial structure using a radar-based metric to assess the performance of deterministic and stochastic models. The reliability of ensemble spreads is also examined. The ensemble of simulations provides uncertainty quantification, which is crucial for hydrological applications. Additionally, we illustrate the methodology with two case studies of daily precipitation fields, showing ensemble means and conditional simulations.

### 4.1 Evaluation of ensemble means

Leave-one-cluster-out cross-validation evaluates the accuracy of ensemble means for all rain gauge stations in the Cevennes region. We compute the mean error and mean absolute error on intense precipitation ($\geq 50$ mm) for the 786 events.

SPAZM has lower spatial interpolation skills than rgISO, rgANISO, arISO, and arANISO (Fig.2). Notably, SPAZM systematically underestimates intense precipitation in the northernmost high-altitude areas, with mean errors ranging from -30 to -50 %, whereas other models (rgISO, rgANISO, arISO, arANISO) show less severe underestimation (-20 to -30 %). SPAZM simultaneously underestimates and overestimates precipitation at neighboring stations, raising concerns about its robustness. rgISO, rgANISO, arISO, and arANISO all underestimate intense precipitation on average. The anisotropy contribution is only visible in mean absolute error for the most northeasterly rain gauge stations.





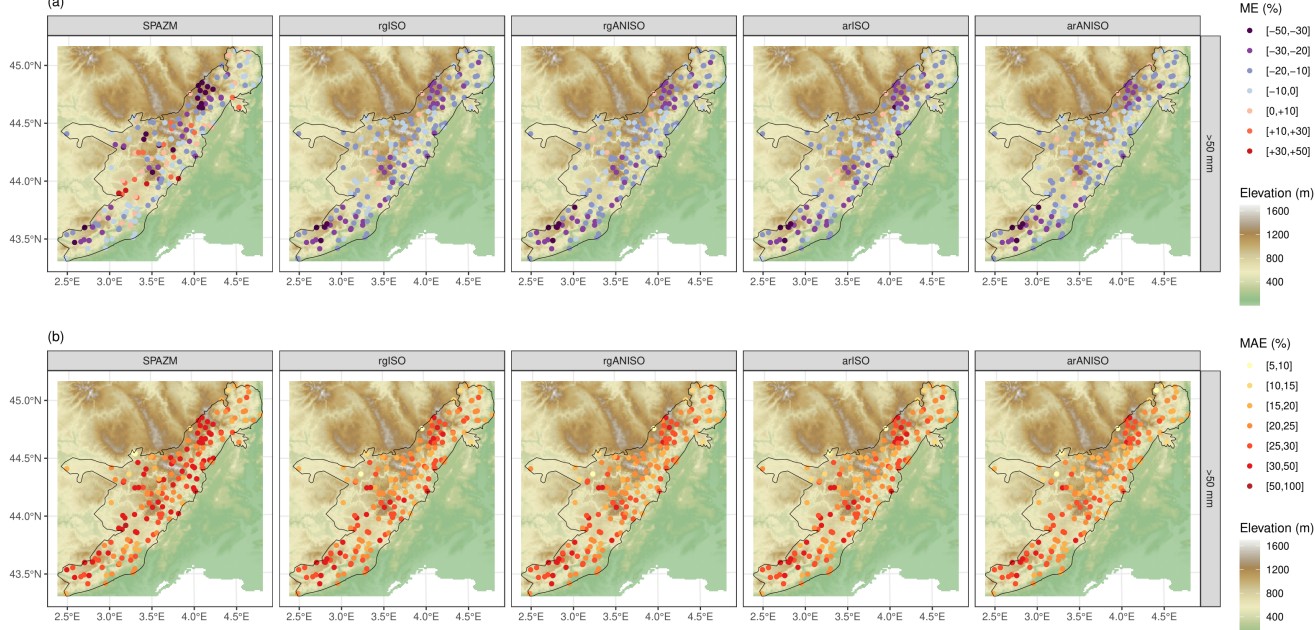

**Figure 2.** Cross-validation mean error (ME, **(a)**) and mean absolute error (MAE, **(b)**) for rain gauge stations in the Cevennes region, considering days with more than 50 mm. Results are shown for deterministic models (SPAZM) and stochastic models (rgISO, rgANISO, arISO, arANISO). For stochastic models, the cross-validation is conducted on the mean of 100 ensemble members from conditional simulations. Black contours indicate the Cevennes region.

## 4.2 Spatial evaluation using radar-analysis

To complement cross-validation, the TWS investigates the image gradient similarities, taking COMEPHORE fields as references. Lower TWS values indicate a better resemblance to COMEPHORE. Figure 3 illustrates the distribution of TWS

values for three event classes: (1) Low anisotropy events ($0.75 \leq \eta \leq 1$), (2) Medium anisotropy events ($0.5 \leq \eta \leq 0.75$), (3) Strong anisotropy events ($0 \leq \eta \leq 0.5$). $\eta$ results from the anisotropy ratio estimation with the daily AROME fields. TWS scores demonstrate a lower gradient image resemblance to COMEPHORE for SPAZM and a better resemblance for arANISO when strong anisotropy occurs. rgANISO does not outperform rgISO in image gradient similarities. The estimation of anisotropic variograms with rain gauges causes a loss of robustness, which is not attenuated even when strong anisotropy

arises. COMEPHORE relies on kriging for stratiform precipitation and radar data for convective precipitation. For this reason, models (rgISO, rgANISO, arISO, arANISO) built on the Trans-Gaussian framework unsurprisingly outperform SPAZM. However, COMEPHORE uses isotropic covariance in the kriging step, meaning only radar carries anisotropy. This suggests that the anisotropic variogram allows the mimicking of radar fields.





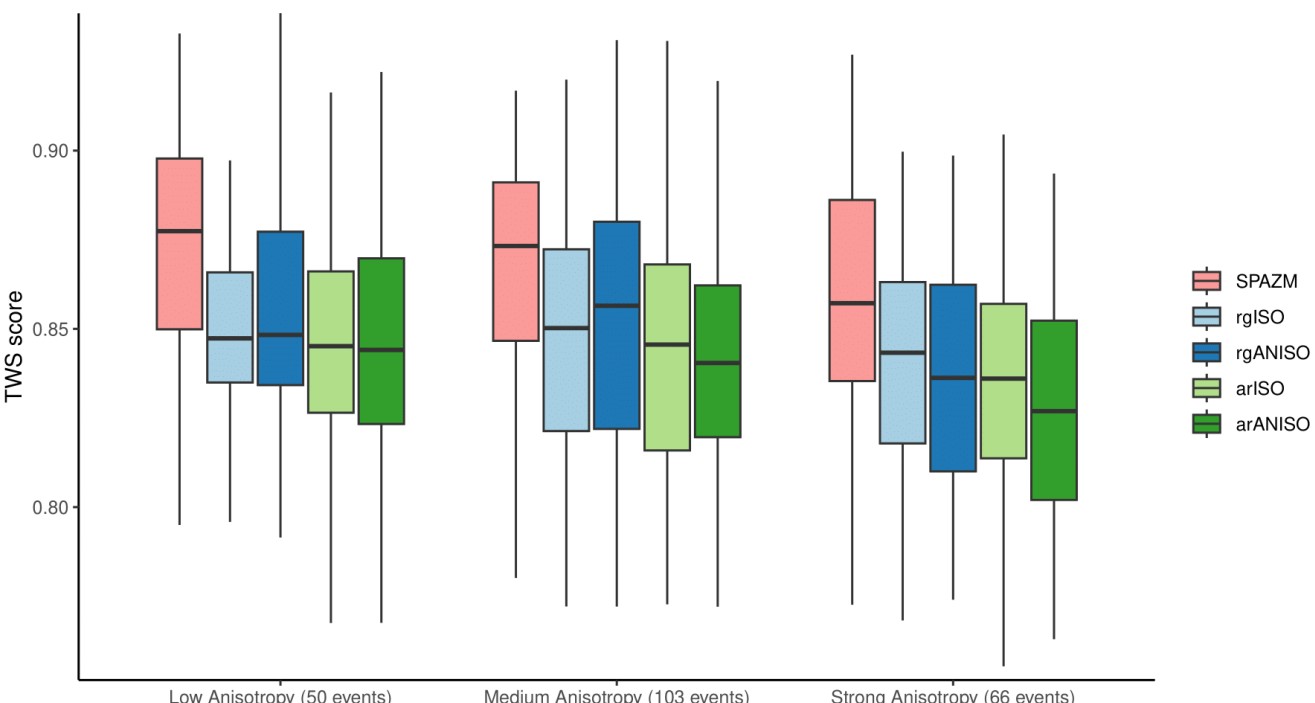

**Figure 3.** Distribution of TWS scores for deterministic models (SPAZM) and stochastic models (rgISO, rgANISO, arISO, arANISO). Low values of TWS scores indicate better image gradients, using COMEPHORE fields as references. The distributions of TWS scores are plotted for low estimated anisotropy ratio ($0.75 \leq \eta \leq 1$, left), medium estimated anisotropy ratio ($0.5 \leq \eta \leq 0.75$, center), and strong estimated anisotropy ratio ($0 \leq \eta \leq 0.55$, right). The number of events is displayed per event class. For the stochastic models, the evaluation is conducted over the mean of 100 ensemble members of conditional simulations.

## 4.3 Probabilistic performances and Uncertainty Quantification

To assess the reliability of ensemble simulations, we apply the same cross-validation approach used for evaluating ensemble means (see Subsection 4.1). The Continuous Ranked Probability Score (CRPS) is computed to quantify probabilistic prediction accuracy, where lower values indicate better performance. Distribution of CRPS scores reveals similar performances for rgISO, rgANISO, and arISO but a notable improvement with arANISO (Fig.4). The error reduction is substantial for strong anisotropy events. This suggests that integrating anisotropy into the covariance structure using AROME enhances the ability of the en-

semble to capture observed daily precipitation. For this reason, we later quantify precipitation uncertainty using conditional simulations from anisotropic covariance.

For the 20 most intense events, the mean catchment precipitations derived from the ensembles of 100-member arANISO from conditional simulations closely agree with COMEPHORE estimates (Fig.5) with a high rain gauge density. The interquartile range of the simulations frequently contains the mean COMEPHORE precipitation for the Chassezac at Sainte-Marguerite







**Figure 4.** Distribution of cross-validation CRPS scores for stochastic models (rgISO, rgANISO, arISO, arANISO). 1 mean value of CRPS is obtained for each station within the Cevennes region for the days with more than 50 mm. Lower CRPS values indicate better spatial interpolation. The distributions of CRPS scores are plotted for low estimated anisotropy ratio ($0.75 \leq \eta \leq 1$, left), medium estimated anisotropy ratio ($0.5 \leq \eta \leq 0.75$, center), and strong estimated anisotropy ratio ($0 \leq \eta \leq 0.5$, right).

and the Thines at Pont de Gournier catchments. For the Chassezac at Pont du Mas catchment, the interquartile range does not necessarily encompass COMEPHORE precipitation, but the full range of the ensemble does encompass it. This suggests the reliability of the uncertainty estimates provided by the ensemble. arANISO is robust, as conditional simulations nearly agree with COMEPHORE estimates in an ungauged scenario. The artificial decrease of rain gauge density causes a slight negative bias in the conditional simulations. The latter frequently contain COMEPHORE precipitations through wider confidence intervals.

As expected, uncertainty increases with precipitation intensity, decreases with catchment size and rain gauge density, and provides asymmetrical distributions on mean precipitations. For instance, on 2014-09-18, the mean rainfall on the small Thines

at Pont de Gournier catchment ranges from 175 mm to more than 450 mm with the complete set of rain gauges, representing
a factor higher than 2.5. In contrast, the uncertainty is lower (factor of 1.33) for larger catchments such as the Chassezac at
265 Sainte-Marguerite.



**Figure 5.** Scatter plot between mean catchment COMEPHORE (x-axis) and simulated (y-axis) precipitations in two rain gauge density
scenario: using all rain gauge available (*Gauged*, left panels), removing the rain gauge of the main catchment (*Ungauged*, right panels). The
20 days with the highest COMEPHORE precipitation are considered for each catchment represented in the vertical panels. The simulations
are derived from conditional simulations with anisotropic covariance (arANISO). The vertical lines in red (grey) illustrate the 25-75 (0-100)
% range of simulations. Black points indicate the ensemble means. The dashed line represents the diagonal.





## 4.4 Case studies

To further illustrate model performance, we analyze precipitation fields for two selected events, using COMEPHORE as the reference. Subsection 2.2.1 describes the COMEPHORE data. Figure 6 presents daily precipitation maps for four catchments. While all models broadly reproduce the spatial structure of COMEPHORE fields, important differences emerge. SPAZM
exhibits both local overestimation and underestimation, producing overly broad precipitation cells. rgISO also yields overly smooth fields, leading to high-intensity underestimation. arISO slightly mitigates this issue, as seen in the 2014-09-18 case, where it captures more intense precipitation than rgISO. rgANISO captures the radar anisotropy for the 2008-05-26 events but produces an inaccurate spatial structure for the 2014-09-18 event. arANISO provides the most realistic interpolation, reducing both excessive smoothing and underestimation of high intensities. It is the only stochastic model able to capture the highest
precipitation values (388–444 mm) in the 2014-09-18 event, demonstrating the advantage of incorporating directional effects.

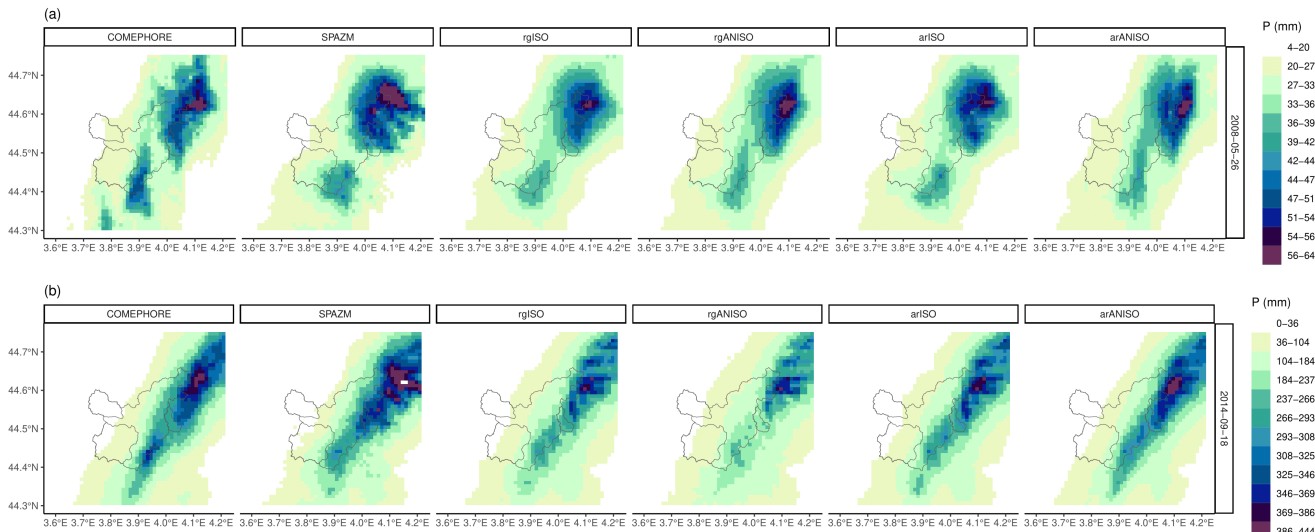

**Figure 6.** Daily precipitation field illustrations, with a focus on the four catchments for the 2008-05-26 **(a)** and 2014-09-18 **(b)** days. Daily precipitation fields are obtained with deterministic models (COMEPHORE, SPAZM) and stochastic models (rgISO, rgANISO, arISO, arANISO). For the stochastic models, the fields correspond to the mean of 100 ensemble members derived from conditional simulations. Precipitation is expressed in mm. Figure 1 shows the locations of the catchments.

A single precipitation field does not reflect the interpolation uncertainty, leading to unwanted smoothing spatial patterns. Conditional simulations provide an attractive alternative to derive a set of equiprobably plausible fields. Figure 7 illustrates five conditional simulations with anisotropic covariance (arANISO) for the 2008-05-26 and 2014-09-18 days, revealing different spatial patterns and precipitation intensities. The Thines at Pont de Gournier catchment, located near the center of the strong
cell intensities, has a substantial mean precipitation uncertainty on 2014-09-18. Simulation 49 gives rise to nearly 250 mm mean catchment precipitation, compared to more than 350 mm for simulation 15.





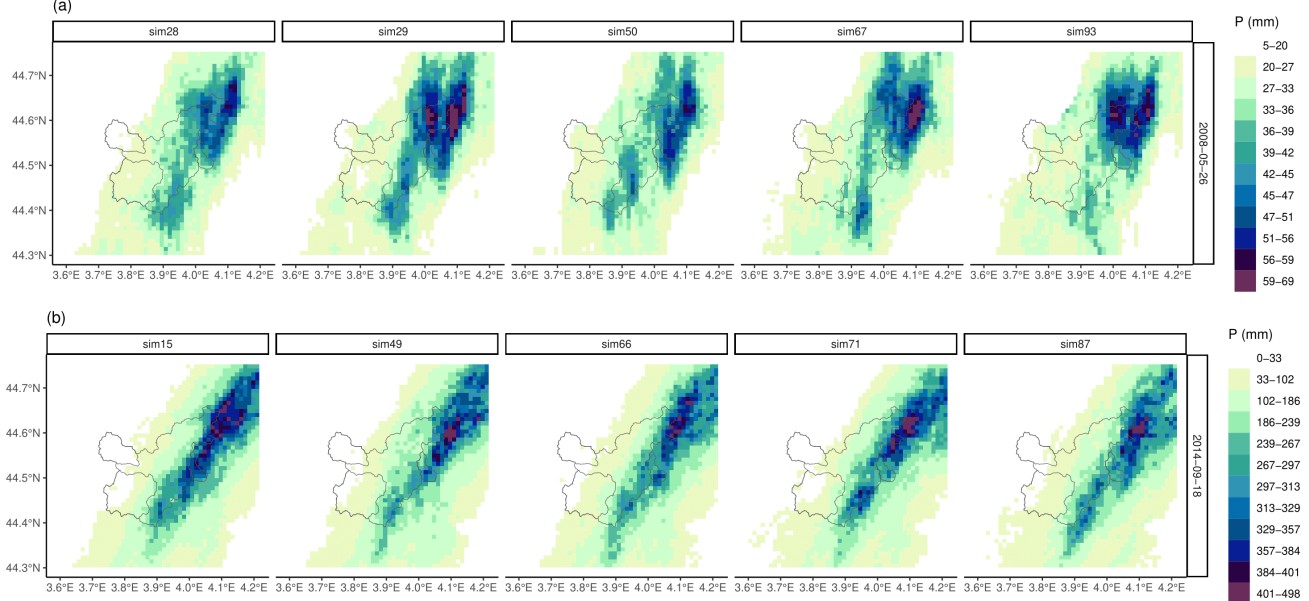

**Figure 7.** Anisotropic (arANISO) illustration of conditional simulations of daily precipitation fields, with a focus on the four catchments for the 2008-05-26 **(a)** and 2014-09-18 **(b)** days. Five members are randomly sampled from a 100 ensemble. Precipitation is expressed in mm. Figure 1 shows the locations of the catchments.

Figure 8 displays the AROME fields and the corresponding anisotropic fitted variograms for the two case studies. Although the AROME fields do not show the same spatial variability and precipitation intensity as COMEPHORE (Fig.6), they include nearly the same anisotropy. The 2008-05-26 event shows a higher spatial correlation in the north-south direction. The 2014-09-18 one is characteristic of a Cevenol episode with a higher spatial correlation in the southwest-to-northeast direction. Table 3 collects information about $\eta$ and $\theta$ estimated in the 786 events. 50 % of events have $\eta < 0.5$. The directional effects mainly occur in the north-south direction to the southwest-to-northeast direction.

| Parameter/Quantile | q0 | q10 | q20 | q30 | q40 | q50 | q60 | q70 | q80 | q90 | q100 |
|---|---|---|---|---|---|---|---|---|---|---|---|
| $\theta$ | 0 | 6 | 11 | 18 | 23 | 29 | 39 | 68 | 158 | 171 | 180 |
| $\eta$ | 1 | 0.80 | 0.67 | 0.62 | 0.56 | 0.50 | 0.48 | 0.43 | 0.38 | 0.31 | 0.18 |

**Table 3.** Anisotropy parameters estimated with daily AROME precipitation fields, namely the angle ($\theta$) of dominant spatial autocorrelation and the anisotropy ratio ($\eta$) with the corresponding orthogonal direction. Deciles summarize the parameter values.

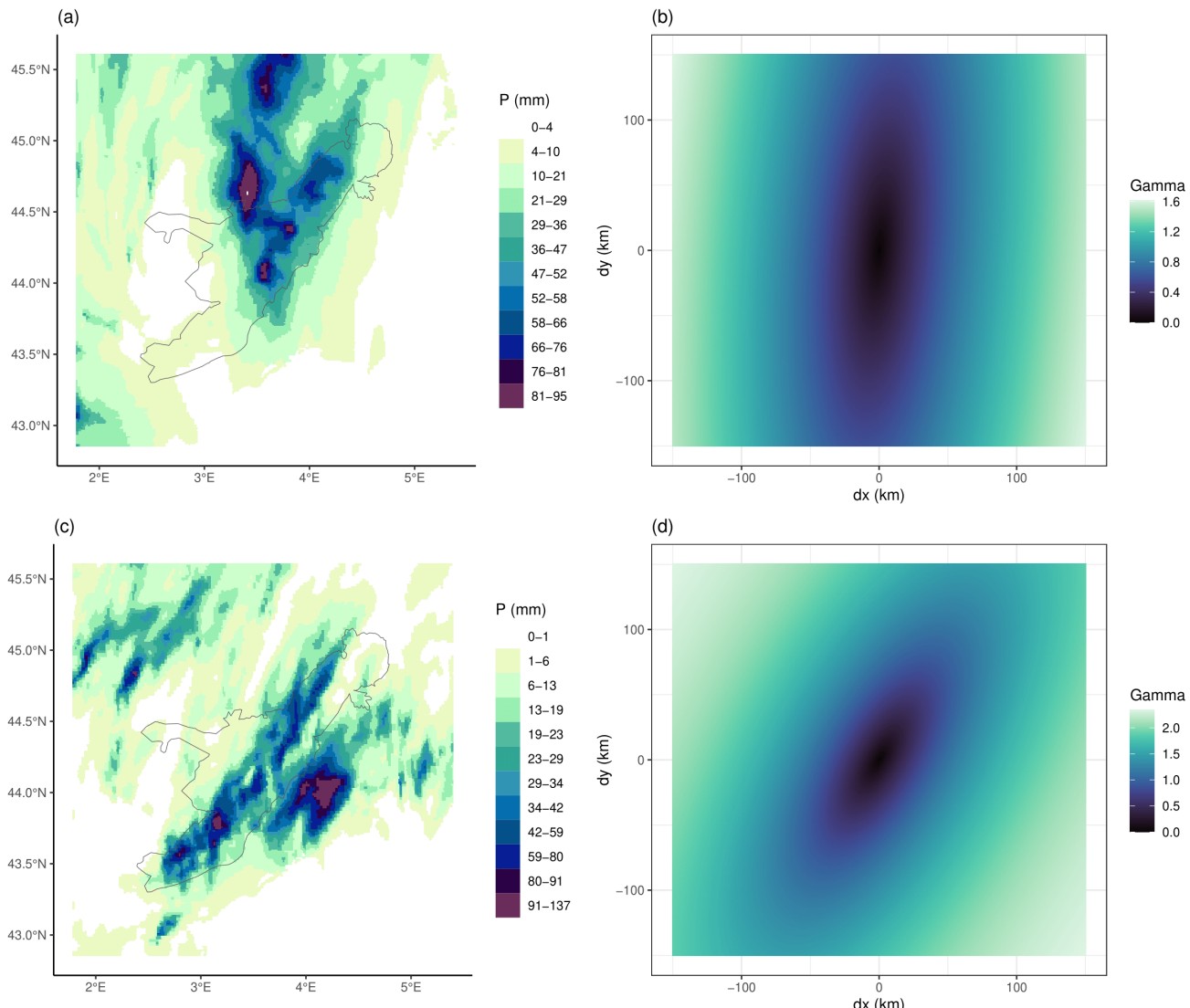

**Figure 8.** Examples of two AROME daily precipitation fields used to derive anisotropic variograms for the 2008-05-26 **(a)** and 2014-09-18 **(c)** days. Precipitation is expressed in mm. Black contours represent the Cevennes region. In the 2D variograms **(b), (d)**, low (high) gamma values indicate strong (weak) spatial autocorrelation between observations, spaced from dx (dy) in horizontal (vertical) km scale.

## 5 Discussion

In this study, we derive anisotropic variograms from CP-RCM simulations for spatial interpolation of precipitation, an approach that, to our knowledge, has not been previously explored. Estimating anisotropy parameters based solely on rain gauge stations is challenging due to the sparse and irregular station networks. However, CP-RCM simulations provide a way of characterizing



anisotropy. We demonstrate that anisotropic variograms improve probabilistic precipitation analysis compared to isotropic ones through both point-based and spatial evaluations. In the following, we discuss the advantages and limitations of our approach and outline the further developments that are needed.

### 5.1 Ensemble from conditional simulations versus local regression approach

This study highlights that Trans-Gaussian Random Fields (kriging with external drift) outperform local regression models in robustness and spatial variability. SPAZM, a local regression interpolator, excessively spreads precipitation cells and overcorrelates precipitation with altitude. This overcorrelation results in poor performance for convective events, leading to regional inconsistencies where SPAZM significantly underestimates or overestimates intense precipitation in localized areas. In contrast, the ensemble mean from Trans-Gaussian Random Fields shows better regional coherence, despite the global underestimation of intense precipitation. This bias is well-known in kriging studies (Hiebl and Frei, 2018). Conditional simulations from Trans-Gaussian Random Fields provide an alternative to generate more realistic precipitation. This approach accurately captures large-scale convective events. Further evaluation is needed in ungauged areas with localized convection cells and complex topography, where orographic effects drive uncertainty.

### 5.2 Covariance estimation

Estimating anisotropic parameters typically requires a high density of rain gauge stations, a condition rarely met in practice. Our results show that CP-RCM simulations, despite their imperfections, effectively capture anisotropic precipitation structures, making them valuable for informing variogram parameters in regions with limited observational data.

In this study, we estimate four covariance models using weighted least squares: (1) isotropic covariance from rain gauge stations (rgISO), (2) anisotropic covariance from rain gauge stations (rgANISO), (3) isotropic covariance from CP-RCM simulations (arISO), and (4) anisotropic covariance from CP-RCM simulations (arANISO). Considering anisotropy into the covariance structure using rain gauges (rgANISO) does not necessarily improve interpolation performances compared to isotropic covariance (rgISO). This result agrees with the few studies on covariance comparisons (Haberlandt, 2007; Shi et al., 2007; Haylock et al., 2008). Using a limited number of rain gauges generates a loss of robustness in estimating the covariance parameters despite actual anisotropy presented in the daily precipitation fields. arISO shows similar performances to rgISO. This suggests that the rain gauge density is high enough to capture spatial autocorrelation in the study region. This also implies that CP-RCM simulations could serve as a viable alternative for estimating spatial autocorrelation in areas with sparse rain gauge station coverage. Estimating anisotropic parameters usually requires a large number of rain gauge stations, which is rarely met in practice. arANISO outperforms the other stochastic models. This approach aligns with previous studies that have used radar data to infer anisotropy through nonparametric estimation (Velasco-Forero et al., 2009; Schiemann et al., 2011; Gyasi-Agyei, 2016). High-quality CP-RCM simulations, increasingly available worldwide, might correctly carry anisotropic information.

The covariance modeling in this study has two main limitations. Firstly, CP-RCM simulations may miss or inaccurately generate precipitation cells, leading to biases in the estimation of covariance parameters such as the range, the anisotropy ratio, and the anisotropy angle. Secondly, the assumption of second-order stationarity may not hold on large spatial domains.



Topographical effects might induce spatial non-stationarity in the covariance structure. The windward mountainsides might encounter more spatial variability than the leeward ones. The study region has daily oceanic or Mediterranean influences, but not both simultaneously, limiting spatial non-stationarity. Several authors implement spatial non-stationarity of the covariance (Paciorek and Schervish, 2006; Risser and Calder, 2017; Risser et al., 2019). For computational reasons, we decided not to include non-stationarity.

Despite those limitations, we are confident in the study's transposability to other regions of similar size with high-quality CP-RCM simulations. Furthermore, the scientific community should explore whether the same methodology can be applied using RCM simulations in regions where CP-RCM simulations are unavailable. The methodology might also be applied to real-time interpolation using numerical weather forecasts.

### 5.3 Quantification of interpolation uncertainty

We assess precipitation interpolation uncertainty at the catchment scale, which is relevant for hydrological applications. For very small catchments ($17 \text{ km}^2$), the simulated mean catchment precipitation can vary widely by a factor of 2.5, highlighting substantial uncertainty despite high rain gauge density. Using a gamma distribution to normalize daily observations creates an asymmetrical distribution of mean catchment precipitation, which appears to be physically reasonable. Other authors (Erdin et al., 2012; Frei and Isotta, 2019) frequently used the Box-Cox transformation to normalize data, causing uncertainties in the back-transformation step. Box-Cox transformations can lead to over- or under-normalization, artificially inflating or deflating the uncertainty in spatial interpolation. Standard square root transformation, a special case of Box-Cox (parameter equal to 0.5), often leads to flattening high precipitation values (Erdin et al., 2012), causing underestimation of spatial interpolation uncertainty.

Uncertainty in precipitation analysis mainly arises from precipitation undercatch and spatial interpolation methods.

In this study, rainfall prevails over snowfall, and precipitation undercatch might be considered negligible. However, in snow-dominated regions, precipitation undercatch is significant (Sevruk et al., 2009; Pollock et al., 2018) under windy conditions. We propose fitting an asymmetrical gamma distribution to the catch ratio to handle this issue. The parameters of this distribution would depend on several factors, including the precipitation phase (rain or snow), precipitation intensity, rain gauge type, its exposure, and wind speed.

While we modeled the primary sources of uncertainty related to spatial interpolation, additional uncertainty remains unaddressed:

- We assume fixed parameters for the gamma distribution used to normalize the data. A natural extension would be to generate multiple sets of gamma parameters and run conditional simulations for each.

- The seasonal climatological background fields used in this study are treated as deterministic despite inherent uncertainties. Further developments should explore probabilistic background fields to propagate uncertainty better.





- In addition, we did not consider the uncertainty related to the estimation of the covariance parameters. Bayesian inference to estimate covariance parameters (Frei and Isotta, 2019), using informative priors derived from CP-RCM simulations, is a logical extension.

Bayesian hierarchical models present an ideal framework to deal with this combination of interdependent uncertainties that can be propagated into hydrological models.

## 6   Conclusion

This study compares different covariance estimations for conditional simulations of daily precipitation using rain gauge stations. The evaluated models include (1) local regression interpolator called SPAZM; Trans-Gaussian Random Fields with (2)
isotropic covariance estimated with rain gauges, (3) anisotropic covariance estimated with rain gauges, (4) isotropic and (5) anisotropic covariances estimated from daily CP-RCM simulations. The expectation of the Random Fields is a function of topographical variables and a seasonal climatological background field. We assess interpolation performance through cross-validation and a spatial metric, using radar-derived analysis as a reference.

Results indicate that Trans-Gaussian Random Fields modeling, whatever the covariance used, consistently outperforms the
local regression interpolator SPAZM in cross-validation and spatial metrics. The geostatistical models are more robust to low rain gauge density and resemble more radar fields. Among the covariance models tested, anisotropic covariance derived from CP-RCM simulations better captures directional precipitation structures observed in radar data and shows superior cross-validation scores for both ensemble mean and spread. By bypassing the lack of robustness of anisotropy estimation using sparse rain gauge networks, this approach reveals the clear value of incorporating anisotropy into the spatial interpolation of
daily precipitation. The obtained ensemble of 100 mean catchment precipitation simulations successfully encompasses radar-based estimates for intense precipitation events, highlighting its potential for uncertainty quantification, a key consideration for hydrological modeling.

These findings suggest that deriving anisotropic variograms from high-quality CP-RCM simulations is a promising approach for probabilistic precipitation interpolation. A natural extension of this work is to integrate the generated precipitation ensem-
bles into probabilistic hydrological modeling, further improving flood risk assessment and water resource management.

## Acknowledgements

We are grateful for the AROME simulations provided by Cécile Caillaud and Diego Monteiro.

## Conflict of interest

The authors declare no conflict of interest.



**Code and data availability**

AROME is available from the Med CORDEX Portal (https://www.medcordex.eu/search/index.php, last access: 02$^{nd}$ May 2024). The Ⓡ codes used to perform the analysis are available upon reasonable request by contacting the first author directly.

**Author contributions**

VD conceived the idea,carried out the analysis and wrote the article. GE, DP, and AF participated in the discussion and design
of this study and contributed to writing and editing the paper.

**Competing interests**

The authors declare no conflict of interest.

**Acknowledgements**

**Financial support**

This research has been supported by Electricité de France (EDF).



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
