# Peer review of "Improving Precipitation Interpolation Using Anisotropic Variograms Derived from Convection-Permitting Regional Climate Model Simulations"

_EGUsphere, 2025_

## Author Comment (AC1)

**Reviewer #1**

**RC1.1 This paper presents a method to interpolate daily (gauges) precipitation data using variograms derived from climate model simulations. The manuscript is well written, and results show potential in the proposed methodology to improve daily precipitation estimations, when gridded rainfall fields such radar-based rainfall estimations may not be available.**

We thank the reviewer for this positive feedback.

**RC1.2 In addition of the methods of validation presented by the authors, I suggest adding a comprehensive comparison between the anisotropic variograms derived from CP-RCM (the target of this paper) with those from the radar-derived precipitation analyses. See more details below. This additional comparison targets directly the approach presented in the manuscript and may provide evidence of the advantages and limitations of the proposed technique.**

Thank you very much for this suggestion. We will compare anisotropic parameters between radar and CP-RCM fields in the sub-section 4.4.

Additional comments:

**RC1.3 L134: Please describe what a Trans-Gaussian Random Field is, as I believe this is the first time the reader is introduced to this term.**

Thank you for this recommendation. We introduce Trans-Gaussian Random Field L134: "To handle the strong non-Gaussianity of daily precipitation (positivity and skewness), we assume that the rainfall field can be represented as a Trans-Gaussian Random Field. That is, there exists a transformation such that the transformed field follows a second-order stationary Gaussian random field.".

**RC1.4 L154: Please elaborate why only 25% of AROME grid cell were selected to calculate the variograms. Were the selected cells from AROME used as 'virtual' gauges to calculate the variograms? Velasco-Forero et al 2009 and other papers describe methods to estimate 2D variograms using all the grid points from radar images that could be applicable to estimate variograms from AROME and COMEPHORE datasets.**

We agree with the reviewer that this part needs clarification. We selected only 25% of AROME grid cells to compute variograms in order to balance spatial representativeness and computational cost. These selected cells were treated as virtual gauges when computing empirical variograms. To confirm robustness, we performed a sensitivity test using 100% of cells for a subset of events (5 first events) , which produced similar

variogram parameter estimates.

[Figure]

Non-parametric 2D variograms can be indeed estimated with all the grid points using FFT. We tried this approach in a first instance in another region, but the obtained covariance matrices  were not semi-positive definite, which is needed for geostatistic applications.

Here are the steps we performed and the diagnostics.

[Figure]

**Step 1** (optional): Normalizing 2D spatial data, removing the mean and dividing by the standard deviation.

**Step 2** : 0-padding of the data.

**Step 3**: Apply FFT.

**Step 4**: Multiply the FFT result by its conjugate to get the spectral density matrix.

**Step 5:** Check that all of the values are positive. This is the case.

**Step 6**: Divide each value by the sum of the spectral densities to sum to 1.

**Step 7**: Take the inverse FFT.

**Step 8**: Take its real components.

**Step 8**: Shift the table.

**Step 9**: Take the central part to remove padding.

We add L154: "We selected only 25% of AROME grid cells to compute variograms in order to balance spatial representativeness and computational cost. These selected cells were treated as virtual gauges when computing empirical variograms. To confirm robustness, we performed a sensitivity test using 100% of cells for a subset of events , which produced similar anisotropy parameter estimates (not shown)."

**RC1.5 L195: Authors are using TWS to verify the spatial structure of the precipitation fields, however spatial multi-scale dependencies are key characteristics of any rainfall fields and authors should add comparisons to account these effects. Seppo Pulkkinen et al. 2019 presents some examples on how to evaluate different rainfall fields based on their multi-scale characteristics (for example figure 8) GMD - Pysteps: an open-source Python library for probabilistic precipitation nowcasting (v1.0)**

We thank the reviewer very much for this recommendation. We compare the slopes of the radius-averaged spectral density, using COMEPHORE data (reference) and the other gridded precipitation products.

[Figure]

We add L201: "Additionally, spectral analysis allows comparison of spatial variability across scales and is well suited to assess whether predictions preserve the multiscale structure of precipitation. The two-dimensional Fourier power spectrum was computed for the reference (COMEPHORE) and predicted fields and averaged radially in wave-number. For scaling processes, the radius-averaged spectrum follows a power-law relationship : $E(k) \propto k^{-\beta}$, where k is the wave number and $\beta$ is the spectral slope. The slope $\beta$, estimated from a linear fit in log–log space, was used as the comparison metric. Similar values of $\beta$ indicate similar spatial variability between reference and predicted fields. We compare the slopes using ME, RMSE, and slope best-fit line metrics. Whereas TWS evaluates short-range gradient image similarities, this comparison evaluates spatial multi-scale dependencies of the daily precipitation fields."

We add L44: "Figure 4 presents the spectral slopes of the reference and predicted precipitation fields. SPAZM systematically underestimates the highest slopes, indicating an overestimation of fine-scale spatial variability. Compared to rgISO, rgANISO degrades the representation of spatial variability for weakly anisotropic events and provides only limited improvement for strongly anisotropic events. In contrast, arISO and arANISO improve the representation of spatial variability, particularly for strongly anisotropic events. Among them, arANISO better reproduces spatial variability, as the highest spectral slopes are no longer underestimated."

**RC1.6 L215: Please indicate where "the ensemble means, a sample of conditional simulations, …" are show (Figure, section???)**

We add L215: "In subsection 4.4, we show ensemble means … ".

**RC1.7 Figure 2: It is hard to discriminate the ME and MAE colours from the topographical background. Please try to use contours for the topography, so score colours become more visible. For the discussion of results of this figure (ME, MAE) please consider if a scatterplot between elevation and scores could help to support your conclusions. If elevation is not relevant here, then please consider removing the topography of the figure.**

We thank the reviewer for this graphical advice. Elevation is not relevant here, so we removed the topography of the figure.

[Figure]

**RC1.8 L235: It is not true that "rgANISO does not outperform rgISO in gauge gradient similarities" as rgANISO TWS score values are mostly lower that rgISO values for the 66 events with strong anisotropy as shown in Figure 3. Also Figure 3 shows that arANISO generally outperforms rgANISO and arISO also generally outperformes rgISO, which could highlight the advantages of using AROME fields to estimate the spatial variability of the rainfall fields.**

We thank the reviewer for pointing out this lack of precision. We replace L237-243 by:

"TWS scores show that SPAZM exhibits lower image gradient similarity to COMEPHORE than all geostatistical approaches, regardless of the event type, indicating a poorer representation of spatial gradients.

For both isotropic and anisotropic configurations, AROME-based methods (arISO and arANISO) consistently outperform their rain-gauge-only counterparts (rgISO and rgANISO). This systematic improvement suggests that the use of AROME fields allows a more accurate estimation of the precipitation covariance structure and, consequently, a better reproduction of spatial gradients.

When considering rain-gauge-based methods only, rgANISO outperforms rgISO for strongly anisotropic events, whereas the opposite behavior is observed for weakly anisotropic events. This indicates that accounting for anisotropy can improve gradient similarity when anisotropy is pronounced, but the limited robustness of anisotropic variogram estimation from rain gauges alone prevents drawing firm conclusions for weakly anisotropic cases.

In contrast, arANISO consistently outperforms arISO, with a marked improvement for strongly anisotropic events and a slight but systematic improvement for weakly anisotropic ones. As no robustness issues arise in the estimation of anisotropic variograms when using AROME fields, these results provide stronger evidence of the

added value of explicitly accounting for anisotropy in the representation of precipitation spatial variability.

COMEPHORE relies on kriging for stratiform precipitation and radar observations for convective precipitation. While COMEPHORE uses isotropic covariance in the kriging step, anisotropy is introduced through the radar component. This suggests that anisotropic variograms, particularly when informed by AROME fields, help better reproduce the spatial structure of radar-derived precipitation patterns."

**RC1.9 L251: last sentence should indicate with dataset is used to estimate the anisotropic covariance. Is from AROME?**

We agree with the reviewer that this information was missing. We add L251 " For this reason, we later quantify precipitation uncertainty using conditional simulations from anisotropic covariance, derived from AROME simulations."

**RC1.10 Figure 5: Please highlight the data points from the event (2014-09-18) in the scatter plots?**

**RC1.11 Figure 5: Please add to the boxes with the catchments name, the code ID and areas of the catchments as described in Table 1.**

**RC1.12 Figure 5: Consider adding best-fit lines fitted across the ensemble mean points on each scatter plot as they could help to illustrate biases in the simulations for each catchment.**

Thank you for these three suggestions that will improve the readiness of this figure.

[Figure]

**RC1.13 Table 3: it would be valuable to add the same stats for COMEPHORE in this table. This allows a direct comparison of the anisotropy parameters derived from AROME and from COMEPHORE.**

Thank you very much for this suggestion. We propose to replace Table 3 by this new figure for a clearer comparison.

[Figure]

We replace L285-287 by : "Figure 8 summarizes the anisotropy parameters η (anisotropy ratio) and θ (anisotropy angle) estimated from AROME and COMEPHORE.

Both datasets show similar preferred anisotropy directions, predominantly oriented south–north (S–N) and southwest–northeast (SW–NE), with AROME exhibiting a slightly stronger S–N component. In both cases, the anisotropy is generally more pronounced along the SW–NE direction."

**RC1.14 Figure 6: please add AROME rainfall fields to this figure as the variogram for arISO and arANISO were derived from AROME. Please consider adding the TWS values for each field.**

We thank the reviewer for this suggestion. TWS will quantify the visual differences that can be seen.

[Figure]

**RC1.15 Figure 7 and discussion. Given that the relatively small size of the catchments with the whole domain, would be valuable to present the distribution of the rainfall values of few (all?) members for each catchment as complement to the precipitation fields?**

Thank you for this graphical suggestion. For better visualisation, we only provide the distribution of the rainfall values for the selected simulation numbers. It helps quantify the precipitation uncertainty.

[Figure]

We replace L280-281: "Simulation 49 gives rise to nearly 250 mm mean catchment precipitation, compared to more than 350 mm for simulation 15" by "Simulation 55 gives rise to nearly 200 mm mean catchment precipitation, compared to more than 500 mm for simulation 95".

#RC1.16 L312: please elaborate when rgANISO improves rgISO and when does not and why?

Thank you for this suggestion. We add L314: "More specifically, for both point cross-validation and spatial variability evaluation, rgANISO outperforms rgISO during strongly anisotropic events, whereas the opposite behavior is observed for weakly anisotropic events. This indicates that anisotropy can be significant but remains difficult to robustly estimate using a rain-gauge network alone, even when the network is dense."

#RC1.17 L334: Please elaborate what it could be needed to extend this methodology to real-time and sub-daily applications as this study only has assessed daily time scales.

"The methodology could also be extended to real-time, sub-daily interpolation. CP-RCM simulations are not continuously updated, so their replacement by hourly numerical weather forecasts should be investigated. At the daily timescale, timesteps are typically considered independent, but this assumption no longer holds at the hourly scale. To address this, temporal dependence should be incorporated into the model, as done in Sideris et al. (2014) and Frey and Frei (2025). A major limitation is the limited availability of sub-daily rain gauges. One potential solution to bypass this shortcoming  is to disaggregate daily interpolated fields using radar data or numerical weather forecasts."

---

## Author Comment (AC2)

**Reviewer #2**

**RC2.1:In this paper, the authors propose a geostatistical framework which relies on a high-resolution regional climate model (RCM) to provide information on the precipitation climatology as well as on the anisotropy of the departures of observations from that climatology. The focus of the paper is on documenting, in particular through two case studies, the added value of the RCM for estimating the anisotropy of the covariogram. A statistical analysis of the performance of the method for 786 events is also presented. The experiment is well designed, and allows the authors to assess separately the impact of using an anisotropic covariogram and the impact of estimating the anisotropy using the RCM. A comparison against a reference interpolation method (SPAZM) is also proposed.**

We thank the reviewer for this positive feedback.

**RC2.2: The introduction reads well but fails to mention a relevant paper by Khedhaouiria et al. (2022) published in NPG which proposed a method based on an ensemble of NWP models to estimate the anisotropy of innovations for optimal interpolation of precipitation.**

Thank you for pointing out this missing article. We add L38: "Numerical weather prediction ensembles have also been explored (Khedhaouiria et al., 2022) to infer background error covariances in data assimilation approaches".

**RC2.3: For the section on domain and data, I suggest including a subsection on the study period. The study period is mentioned in the section on COMEPHORE, but it would be simpler to add a section dedicated to the study period after the section on meteorological data, because the study period is constrained by the availability of COMEPHORE and AROME. A subsection on observed data should also be added. The rain gauge network is currently described in the study domain subsection.**

We agree with the reviewer that more subsections are needed in the section on domain and data. We propose to move L69-73 into a separate subsection named "Rain gauge observations". We also include a fourth subsection called "study period" with the following text: "The study period ranges from 1982 to 2018, which corresponds to the availability of AROME simulations. 786 precipitation events (nearly 20 events per year), defined as the days with at least 50 mm recorded at a minimum of five rain gauges, are selected".

**RC2.4: In the sub-section describing the CP-RCM AROME, I would like the authors to provide more details on the model configuration, more specifically w.r.t. to the ability of the system to represent specific events and not only the climatology of precipitation over the domain. This is important since AROME is used later to inform the interpolation method on the anisotropy of the covariogram, but not on the amount of precipitation associated with the event. Figure 4.6 and 4.8 show that there are significant discrepancies between COMEPHORE and AROME precipitation fields. Is this happening because CP-RCM AROME is not sufficiently constrained by ERA-Interim or is**

it inherent to the predictibility of precipitation events in this region? Would we expect a similar degree of agreement for a short-term forecast of precipitation based on AROME? Is ERA-Interim only used as boundary conditions or is some form of spectral nudging used to prevent RCM model drift? How far is the study domain from the ALADIN and AROME boundaries? Given the model configuration, do we expect AROME to only be able to provide information on the precipitation climatology but yet be able to provide useful information on the anisotropy of the precipitation structures? Why would that be the case?

We agree with the reviewer that more details are needed on AROME configuration. We modify the AROME subsection:

"AROME simulations (Caillaud et al., 2021) are produced with the convection-permitting RCM AROME in its NWP configuration cycle 41t1, which uses 60 vertical levels from 10 m to 1 hPa, including 21 levels below 2000 m to better resolve the lower-tropospheric dynamics over complex Alpine terrain. In this CP-RCM configuration, deep convection is explicitly resolved, while only shallow convection remains parameterized. The AROME domain over the Alpine region lies approximately 300–400 km from the lateral boundaries, which are forced by hourly outputs from the CNRM-ALADIN RCM (Nabat et al., 2020). ALADIN uses 91 vertical levels together with spectral nudging to ensure consistency with the large-scale circulation imposed by the ERA-Interim reanalysis (Dee et al., 2011). AROME simulations are available at the hourly timescale for the Alpine region, as described in the Flagship Pilot Study of the Coordinated Regional Climate Downscaling Experiment (CORDEX-FPS,Fantini et al. (2018)), at 2.5 km spatial resolution, and cover the 1982–2018 years. Hourly outputs are aggregated to a daily timescale.

Previous studies (Ban et al., 2021; Caillaud et al., 2021; Monteiro et al., 2022) show that AROME provides a more realistic representation of intense precipitation than its driving model ALADIN, despite persistent biases. In a Lagrangian evaluation over the Mediterranean region, which includes our study domain, Caillaud et al. (2021) report that AROME simulations reproduce well the location, intensity, frequency, and interannual variability of heavy precipitation events. Remaining biases are mostly due to the model, rather than insufficient constraint from ERA-Interim. In the AROME model, very intense daily amounts (> 200 mm day$^{-1}$) tend to be underestimated, the spatial extent of intense convective cells is overestimated, and their propagation speed is slightly too high. These biases could be reduced by further refining horizontal and vertical resolution and by improving the parameterization of residual shallow and dry convection.

Unlike a short-term NWP forecast, AROME-climate simulations do not assimilate observations such as radar reflectivity; therefore, they are not expected to reproduce individual events exactly, but rather their typical spatial structures. Consequently, even if absolute precipitation amounts may be biased, the spatial organization and anisotropy of intense precipitation systems, key for informing the anisotropic covariogram, may be sufficiently well captured by AROME to support our interpolation framework."

**RC2.5: In the methods section, the authors choose to consider observed precipitation of less than 0.5 mm as zeros for the purpose of normalizing the precipitation field. How was this number chosen? Are results sensitive to this choice? The back-transformation introduces a bias, which the authors do not take into account (see Van Hyfte et al., 2023, Tellus A). Can the authors quantify the impact of ignoring this source of bias on their analysis?**

We thank the reviewer for this remark. 0.5 mm is a standard choice in censoring daily precipitation (e.g Naveau et al. 2016). Results are not very sensitive to this choice.

[Figure]

The above figure highlights the CRPS score with the arANISO model. A too high threshold (1 mm) leads to bias for moderate precipitation (1-5 mm, 5-10 mm), and a too low threshold (0.1 mm) degrades the estimation of 0-1 mm. 0.5 mm corresponds to a good compromise.

We used a Quantile-Quantile mapping to transform and back-transform precipitation. We did not use the traditional Box Cox-transformation, that is a non linear transformation and therefore introduces a bias in the back-transformation step. The uncertainty of our transformation/back-transformation step only lies in the estimation of the gamma parameters. We discuss this idea in the subsection 5.3.

**RC2.6: The authors chose 786 precipitation events to evaluate the proposed method. It would be interesting to know more about the type of events that were selected. Please categorized them by weather regime and season. In particular, can you identify events for which orographic intensification is expected and events for which snow was observed at higher elevations? Do we expect the performance and ranking of the methods to vary depending on the type of event ?**

Thank you for this suggestion. The 786 precipitation events selected in this study are predominantly associated with southerly atmospheric flow and central depression patterns, for which strong orographic intensification is typically expected over the region. During winter, some of these events also include snowfall at high elevations.

To assess whether the performance of the interpolation methods varies with event type, we stratified the 786 events according to the eight weather regimes defined in Garavaglia et al. For each regime, we computed the CRPS. Overall, the relative ranking of the methods is consistent across weather regimes. The rgANISO model provides a better covariance estimation than rgISO for regime 3 (oceanic flow). However, this improvement is not observed for regimes 4 and 7, which correspond to the heaviest precipitation events. Across all regimes, arANISO remains the best-performing model, suggesting that the superiority of this method is robust to meteorological conditions.

We propose to add at L70:
 "Most of the 786 precipitation events arise from southerly atmospheric flow and central depression patterns, where strong orographic intensification is expected. Some winter events also include snowfall at high elevations."

We also include a figure of CRPS by weather regime in the Supplementary Material.

[Figure]

**RC2.7: In the results section, the authors should present and discuss the covariograms that are obtained for each of the two case studies. The 2D covariograms derived from AROME are presented in Figure 4.8, but that does not tell us how well it fits the experimental covariogram. Furthermore, no information is provided on the fitted covariograms for the other three experiments (rgISO, rgANISO and arISO). This is important, in particular to show that the choice of an exponential covariogram is appropriate based on the data. Did the authors check that the exponential variogram provided a good fit for the 786 events considered in this study?**

We thank the reviewer for this comment. We replace Figure 8 by the below figure.

[Figure]

Moreover, we make visual inspections of variogram fitting.

[Figure]

Here are the empirical variogram and the fitted exponential and Matérn variograms for a given day. The exponential variogram fits the empirical one well, benefiting from the large number of estimation pixels used as virtual gauges. However, the Matérn variogram appears to better capture the short-range spatial variability, providing a smoother representation than the exponential variogram. A cross-validation would be needed to assess the best variogram for precipitation interpolation.

**RC2.8: In the discussion, the authors address many limitations of the method, in particular the fact that it would be difficult to apply on a larger domain. This is an important limitation, because it would seem impractical to deploy such a complex interpolation method operationally if it cannot be applied on a large domain. I encourage the authors to propose a workflow that would allow the application of the method on a**

larger domain. Could it not be applied watershed by watershed? Would the cost of doing so be prohibitive? Are there other solutions?

We thank the reviewer for raising this important point. We agree that the applicability of the method over a large domain is a key consideration for operational use, and that the current implementation is best suited for moderately sized regions. Extending the approach to larger and topographically complex domains indeed requires additional methodological considerations.

A first practical solution would be to apply the method watershed by watershed, or more generally to divide the study area into climatically homogeneous sub-regions. This would ensure that the covariance structure remains locally stationary while preserving hydrologically coherent areas. Simulations would remain consistent within each major watershed, which is often sufficient for hydrological applications. The main challenge is the definition of appropriate sub-region boundaries and areas, as overly small regions may lose spatial coherence while overly large regions may violate stationarity assumptions.

A more robust solution for large-scale applications would involve adopting non-stationary covariance models. Some geostatistical approaches include non-stationary covariance: (i) geographical coordinate deformation to map complex terrain into a space where covariance is closer to stationary, or
 (ii) locally stationary covariance models in which parameters evolve spatially but do so smoothly across the domain.
 Such approaches would allow the method to be applied on much larger and more heterogeneous domains without the need to arbitrarily define sub-regions, but with heavier computing times.

We have added the following text at L330:
 "Extending the method to larger and topographically complex domains would require non-stationary covariance. A practical option is to partition the region into climatologically homogeneous sub-regions, ideally preserving major watershed boundaries to maintain hydrological consistency. Alternatively, a more scalable solution is to incorporate non-stationary covariance structures, for example, through geographical coordinate deformation (Youngman, 2023) or locally stationary covariance models (Paciorek and Schervish, 2006; Risser and Calder, 2017), which would allow spatial dependence to evolve smoothly across the domain. These approaches would make the method suitable for operational applications over larger domains."

#RC2.9: Although evaluating the method on significant events (where more than 50mm/day was observed by at least 5 gauges), in practice one would likely want to apply the method for all days. Did the authors assess how well the method performs for less intense precipitation events?

We thank the reviewer for this remark. In this study, we compare covariance estimation on a collection of 786 daily events, and compute evaluation metrics only on the stations exceeding 50 mm/day. However, we agree that in practical applications the method would need to perform well across a full range of precipitation intensities.

We provide in the below figure the metrics for the same set of 786 daily events, including additional precipitation intensity classes (1-20 mm, 20-50 mm). The ranking is similar for the three precipitation classes considered, indicating that the conclusions drawn for intense precipitation events generalize to lighter precipitation.

We also include a figure of CRPS by precipitation intensity class in the Supplementary Material.

[Figure]

#RC2.10: The authors also mention in the discussion the possibility of applying the method using numerical weather forecasts rather than using a RCM. I think this is worth discussing in more details. In particular, I would expect the precipitation field of a

short-term forecast to correlated better with observed precipitation, and thus it might be possible to infer more from the forecast than simply the anisotropy of the precipitation field. Furthermore, one might have access to an ensemble of weather forecasts.

Thank you very much for this remark. We agree that numerical weather forecasts (NWP) should exhibit a higher correlation with observed precipitation than CP-RCM simulations, due to the assimilation of past radar reflectivity. Consequently, it is likely that NWP forecasts could provide not only the anisotropy of the precipitation field, but also information on precipitation intensity and spatial variability. Moreover, the provision of ensemble NWP forecasts would allow us to quantify additional interpolation uncertainty.

We have added at L334:
 "NWP assimilate past radar reflectivity and should therefore display a higher correlation with observations than CP-RCM simulations. As a result, NWP may allow us to extract precipitation intensity, spatial patterns and spatial variability, while quantifying interpolation uncertainty through conditional simulations and the use of ensemble NWP forecasts. A natural follow-up would be to use NWP forecasts as both drift and covariance structures within a kriging-with-external-drift framework (e.g. Velasco 2009, Schiemann 2011)"

**RC2.11: Finally, one important aspect of precipitation interpolation that is not discussed in this paper is the issue of quality control. When interpolating precipitation observations, in particular in complex terrain, the issue of quality control is central because it can be very difficult to identify problematic observations based on neighboring stations, given the impact of orography on precipitation amounts over short distances. I understand that this issue might be out of scope for this paper, but I wonder if it was an issue for the authors when applying the method over 786 events. Was the observed data quality controlled? How? Could the presence of outliers impact the results of your analysis? Could your method be used to improve the quality control process through the use of cross-validation? This would likely be crucial to address before the method can be used for real-time applications.**

We agree with the reviewers that quality control is central before providing gridded precipitation analysis, especially over a long period where rain gauges can change from locations and measurement devices, which can cause temporal discontinuity. Moreover, in complex terrain, strong spatial gradients limit the ability to identify outliers (sensor malfunctions) from neighboring stations.

The observed data was not quality control in this study, which is beyond the scope of this study. Because we do not work with climatological statistics, temporal discontinuity is not a major issue. However, outliers may still occur and could affect the interpolated fields. Such outliers may influence some local results, but we do not expect them to alter the conclusions of this study.

Before applying the method in an operational or real-time context, we will consider a quality control procedure, including a spatial anomaly analysis to identify outliers that exceed physically plausible differences from nearby stations under similar terrain characteristics,  followed by an homogeneity test such as the Standard Normal Homogeneity Test (SNHT; Alexandersson 1986). The R package climatol encompasses those corrections.

Moreover, missing values are commonly filled using a linear regression with nearby rain gauges as predictors. Our approach could be used as a substitute of the linear regression, providing a value, and the associated uncertainty.

We propose to add L72: "The observed data was not quality control in this study. Because we do not work with climatological statistics, temporal discontinuity is not a major issue. However, outliers may still occur and could affect the interpolated fields. Such outliers may influence some local results, but we do not expect them to alter the conclusions of this study. Before applying the method in an operational or real-time context, a quality control procedure is needed, including a spatial anomaly analysis to identify outliers that exceed physically plausible differences from nearby stations under similar terrain characteristics,  followed by an homogeneity test such as the Standard Normal Homogeneity Test (SNHT; Alexandersson 1986)".

---

## Author Comment (AC3)

[revised manuscript text omitted]

**2.3.2 AROME**

AROME simulations (Caillaud et al., 2021) are  produced with the convection-permitting RCM AROME in its NWP configuration cycle 41t1, which uses 60 vertical levels from 10 m to 1 hPa, including 21 levels below 2000 m to better resolve the lower-tropospheric dynamics over complex Alpine terrain. In this CP-RCM  configuration, deep convection is explicitly resolved, while only shallow convection remains parameterized. The AROME domain over the Alpine region lies approximately 300–400 km from the lateral boundaries, which are forced by hourly outputs from the CNRM-ALADIN RCM  (Nabat et al., 2020). ALADIN uses 91 vertical levels together with spectral nudging to ensure consistency with the large-scale circulation imposed by the ERA-Interim reanalysis (Dee et al., 2011).  AROME simulations are available at the hourly timescale for the Alpine region, as described in the Flagship Pilot Study of the Coordinated Regional Climate Downscaling Experiment  (CORDEX-FPS, Fantini et al., 2018), at 2.5 km spatial resolution, and cover the  1982–2018 years. Hourly outputs are aggregated to a daily timescale.  Previous studies (Ban et al., 2021; Caillaud et al., 2021; Monteiro et al., 2022)  show that AROME provides  a more realistic representation of intense precipitation  than its driving model ALADIN, despite persistent biases. In a Lagrangian evaluation over the Mediterranean region, which includes our study domain, Caillaud et al. (2021) reports that AROME simulations reproduce well the location, intensity, frequency, and interannual variability of heavy precipitation events. Remaining biases are mostly due to the model, rather than insufficient constraints from ERA-Interim. In the AROME model, very intense daily amounts ($> 200$ mm. day$^{-1}$) tend to

be underrepresented, the spatial extent of intense convective cells is overestimated, and their propagation speed is slightly too high. These biases could be reduced by further refining horizontal and vertical resolution and by improving the parameterization of residual shallow and dry convection. Unlike a short-term NWP forecast, AROME-climate simulations do not assimilate observations such as radar reflectivity; therefore, they are not expected to reproduce individual events exactly, but rather their typical spatial structures. Consequently, even if absolute precipitation amounts may be biased, the spatial organization and anisotropy of intense precipitation systems, key for informing the anisotropic covariogram, may be sufficiently well captured by AROME to support our interpolation framework.

**2.3.3 SPAZM**

SPAZM (Gottardi, 2009) creates daily precipitation analyses at 1 km spatial resolution over a large part of France since 1948 and is continually updated by Electricité de France (EDF) for hydrological modeling. The daily precipitation fields constitute an adjustment of a climatological background field using the daily rain gauge amounts. A climatological background field represents the average daily structure of the precipitation field, where the influence of topography is easier to account for. The background fields incorporate local orographic effects conditioned by eight weather patterns. SPAZM leads to balanced water budgets at the annual scale, even in mountainous areas (Ruelland, 2020). However, the spatial variability of the daily fields is questionable, with an unrealistic correlation to the altitude. Moreover, SPAZM underestimates the spatial variance of intense precipitation, resulting in overly smoothed precipitation fields (Penot, 2014).

**2.4 Study period**

The study period ranges from 1982 to 2018, which corresponds to the availability of AROME simulations. 786 precipitation events (nearly 20 events per year), defined as the days with at least 50 mm recorded at a minimum of five rain gauges, are selected. Most of the 786 precipitation events arise from southerly atmospheric flow and central depression patterns, where strong orographic intensification is expected. Some winter events also include snowfall at high elevations.

**3 Methods**

**3.1 Spatial Interpolation**

This study develops a probabilistic geostatistical framework for daily precipitation interpolation, using four variograms derived from rain gauges and CP-RCM simulations. Unlike traditional approaches that rely solely on rain gauge data, our method exploits CP-RCM fields to estimate daily isotropic and anisotropic covariance structures. In the following, we detail the stochastic modeling and evaluation procedures.

**3.1.1 Data transformation**

150 Geostatistics involving kriging methods are built in the Gaussian paradigm (Diggle et al., 2003). However, daily precipitation data often exhibits a positively skewed distribution due to left truncation at zero precipitation and the presence of high values. To address this issue, we assume the positive precipitation follows a Gamma distribution and apply quantile-quantile transformation (Gyasi-Agyei, 2018) to normalize the data each day. We perform Gamma fitting using the maximum-likelihood approach from the *fitdistr* Ⓡ package (Delignette-Muller and Dutang, 2015). Spatial interpolation is then conducted in the

155 Gaussian domain. Skewness and heteroscedasticity of the precipitation data are thus explicitly considered (Erdin et al., 2012), theoretically resulting in greater interpolation uncertainty in areas with high precipitation.

The modelization steps include:

1. Gamma to Normal transformation of positive precipitation:
   $Y^* = \Phi^{-1}[u_0 + (1 - u_0)F_\gamma(Y)]$, where Y represents daily positive observed precipitations, $F_\gamma$ the distribution function of a Gamma distribution, $\Phi^{-1}$ is the quantile function of a Gaussian distribution with zero mean and unit variance,

2.  $u_0$  is the empirical probability of zero precipitation  estimated by the number of rain gauges with less than 0.5 mm over the total number of rain gauges, and $Y^*$ are the normalized precipitations.

165 3. Gamma to Normal transformation of zero precipitation:
   The zeros are transformed to $\Phi^{-1}[\mathcal{U}(0.01, u_0)]$, $\mathcal{U}$ is the uniform distribution.

4. Conditional simulations from geostatistical models (detailed in subsection 3.1.2).

5. Back-Transformation of precipitation predictions into Gamma distribution:

    $\hat{Y} = \begin{cases} F_\gamma^{-1}\left\{ \frac{\Phi(Y^*) - u_0}{1 - u_0} \right\}, & \Phi(Y^*) \\ 0, & \text{otherwis} \end{cases}$

170 Once the precipitation data is normalized, we model its spatial structure using geostatistical models, focusing on estimating covariance structures through variogram fitting.

**3.1.2 Geostatistic models**

Precipitation should be centered in the first instance to estimate its covariance structure accurately (Schabenberger and Gotway, 2005). To handle the strong non-Gaussianity of daily precipitation (positivity and skewness), we assume that the rainfall field

175 can be represented as a Trans-Gaussian Random Field. That is, there exists a transformation such that the transformed field follows a second-order stationary Gaussian random field. The expectation of the Trans-Gaussian Random Field is modeled in the Gaussian domain as a function of geographical predictors (longitude, latitude, and altitude) and a seasonal climatological

background field. The literature commonly employs geographical predictors such as altitude in kriging with external drift model (Bárdossy and Pegram, 2013). The seasonal climatological background fields have been built through Random Forest modelization using CP-RCM simulations in (Dura et al., 2024). Their construction is based on the idea that the fine-resolution simulations of climate models should summarize the topographical influence on precipitation. Climatological background fields have been used broadly (see e.g. Hunter and Meentemeyer, 2005; Gottardi, 2009; Masson and Frei, 2014) to summarize long-term precipitation patterns influenced by orography, improving the robustness of daily precipitation interpolation. We successfully check the absence of multicollinearity by computing the correlation matrix among the predictors.

Spatial interpolation requires the estimation of a spatial covariance. We fit each day four different exponential variograms, all including a nugget effect:

- an isotropic variogram estimated with rain gauge observations (rgISO),

- an anisotropic variogram estimated with rain gauge observations (rgANISO),

- an isotropic variogram derived from daily AROME precipitation field (arISO),

- an anisotropic variogram derived from daily AROME precipitation field (arANISO).

The nugget parameter corresponds to micro-scale precipitation variability and measurement errors. The exponential choice is standard in numerous geostatistical studies (Masson and Frei, 2014; Frei and Isotta, 2019) because it effectively captures the gradual decline in autocorrelation as the separation distance increases. The weighted least-square estimation of the variograms is done in the Gaussian domain for both rain gauge and AROME precipitations. The weights are proportional to the number of pairs of observations and inversely proportional to the squared average distance between them.  We selected only 25 %  of AROME grid cells  to compute variograms to balance spatial representativeness and computational cost. These selected cells were treated as virtual gauges when computing empirical variograms. To confirm robustness, we performed a sensitivity test using 100 % of cells for a subset of events, which produced similar anisotropy parameter estimates (not shown). We note $\eta$ and $\theta$ as the estimated anisotropy ratio and angle. Schiemann et al. (2011) describes the methodology in more detail, with an ordinary least square estimation in contrast to a weighted least square estimation. The variograms are estimated with the *gstat* ℝ packages (Pebesma, 2004).

**3.1.3 Conditional simulations**

Conditional simulation of Trans-Gaussian Random Field is conducted using sequential Gaussian simulations implemented in the *gstat* ℝ package (Pebesma, 2004) to create each day an ensemble of 100 simulations. Gyasi-Agyei (2018) states that the conditional simulations obtained with *gstat* are too granular. They decided to apply a $3 \times 3$ window smoothing technique. We do not apply the same post-processing step because we will, in future developments, employ mean catchment precipitation, which is not sensitive to the granularity. While this ensemble size is sufficient for representing precipitation variability (Frei and Isotta, 2019), it does not account for all sources of uncertainty, such as covariance parameters inference or precipitation undercatch (see subsection 5.3).

**3.2   Evaluation**

In our analysis, the ensemble mean of the conditional simulations is equivalent to the result obtained using deterministic kriging with external drift (Schabenberger and Gotway, 2005) (KED). Later in the article, we will use the terms "kriging" and "ensemble mean" interchangeably. While kriging captures the most likely precipitation value, the conditional simulations capture the entire ensemble of possible realizations, providing spatial uncertainty. To assess the performance of the spatial interpolation models, we evaluate both deterministic and probabilistic interpolated precipitation fields. The evaluation focuses on three key aspects:

- point-scale accuracy using cross-validation on rain gauge stations ;

- spatial structure by comparing interpolated fields with radar-derived precipitation analyses ;

- uncertainty quantification by analyzing the ensemble spread of conditional simulations at the catchment scale.

The models are evaluated on 786 precipitation events (nearly 20 events per year), defined as the days with at least 50 mm recorded at a minimum of five rain gauges.

**3.2.1   Cross-validation of ensemble means**

The deterministic performance of the spatial interpolation models is evaluated through a leave-cluster-out cross-validation scheme. Unlike traditional leave-one-out cross-validation, this procedure removes entire clusters of neighboring stations to better mimic ungauged conditions and accentuate performance differences between the models. An accurate covariance estimation is crucial for sparse rain gauge station networks. Sequentially, a cluster of neighboring stations is removed from the training dataset and then predicted. Ten groups of stations are created by the K-means algorithm (MacQueen, 1967). The clustering variables are the longitude and the latitude of the stations. Figure 1 indicates the rain gauge station clusters.

The evaluated models include (1) SPAZM as the deterministic interpolation model , (2) rgISO: Kriging with an external drift with an isotropic variogram estimated from rain gauge data, (3) rgANISO: Kriging with an external drift with an anisotropic variogram estimated from rain gauge data, (4) arISO: Kriging with an external drift with an isotropic variogram derived from CP-RCM precipitation fields, (5) arANISO: Kriging with an external drift with an anisotropic variogram derived from CP-RCM precipitation fields. Mean error (ME) measures bias, indicating systematic over- or underestimation. Mean absolute error (MAE) quantifies overall prediction accuracy.

**3.2.2   Spatial Structure Evaluation with Radar Analyses**

Beyond point-scale accuracy, we assess whether interpolated precipitation fields reproduce the true observed spatial variability. The Teweles–Wobus Score (TWS,Teweles and Wobus (1954)) is used to compare the spatial gradients of interpolated precipitation fields with those from the radar-based COMEPHORE fields. Subsection 2.3.1 describes the COMEPHORE data. We consider COMEPHORE fields as a reference in the study domain because of the good radar coverage, allowing an ac-

240 curate representation of the size and direction of the heavy precipitation cells. It should be noted that for stratiform events, COMEPHORE could favor geostatistic models with isotropic covariance by construction.

TWS is defined as:

$$\text{TWS} = \frac{\sum_{(s,s')\in\text{Adj}} |(p_s - p_{s'}) - (\hat{p}_s - \hat{p}_{s'})|}{2\sum_{(s,s')\in\text{Adj}} \max(|p_s - p_{s'}|, |\hat{p}_s - \hat{p}_{s'}|)},$$

where $\text{Adj}$ is the set of adjacent grid points $(s, s')$ in the northern-southern and eastern-western direction within the study domain, $p_s$ ($\hat{p}_s$) is the COMEPHORE (predicted) daily precipitation amount at the grid point $s$ divided by the maximum grid amount. In this part, the interpolated fields are obtained using all rain gauges in the study domain. This metric is particularly

245 relevant for evaluating the performance of variograms in capturing directional precipitation patterns. Lower values of TWS indicate better agreement with radar-based spatial structures.

Additionally, spectral analysis allows comparison of spatial variability across scales and is well-suited to assess whether predictions preserve the multiscale structure of precipitation. The two-dimensional Fourier power spectrum was computed for the reference (COMEPHORE) and predicted fields, and averaged radially in wave-number. For scaling processes, the

250 radius-averaged spectrum follows a power-law relationship: $E(k) \propto k^{-\beta}$, where $k$ is the wave number and $\beta$ is the spectral slope. The slope $\beta$, estimated from a linear fit in log–log space, was used as the comparison metric. Similar values of $\beta$ indicate similar spatial variability between reference and predicted fields. We compare the slopes using ME, RMSE, and slope best-fit line metrics. Whereas TWS evaluates short-range gradient image similarities, this comparison evaluates spatial multi-scale dependencies of the daily precipitation fields.

**3.2.3 Probabilistic Performances and Uncertainty Quantification**

[revised manuscript text omitted]

SPAZM exhibits lower image gradient similarity to COMEPHORE than all geostatistical approaches, regardless of the event type, indicating a poorer representation of spatial gradients. For both isotropic and anisotropic configurations, AROME-based methods (arISO and arANISO) consistently outperform their counterparts that use only rain gage data (rgISO and rgANISO). This systematic improvement suggests that the use of AROME fields allows for a more accurate estimation of the precipitation covariance structure and, consequently, a better reproduction of spatial gradients. Using only rain gauges, rgANISO outperforms rgISO for strongly anisotropic events, whereas the opposite behavior is observed for weakly anisotropic events. This indicates that accounting for anisotropy can improve gradient similarity when anisotropy is pronounced, but the limited robustness of anisotropic variogram estimation from rain gauges alone prevents drawing firm conclusions for weakly anisotropic cases. In contrast, arANISO consistently outperforms arISO, with a marked improvement for strongly anisotropic events and a slight but systematic improvement for weakly anisotropic ones. As no robustness issues arise in the estimation of anisotropic variograms when using AROME fields, these results provide stronger evidence of the added value of explicitly accounting for anisotropy in the representation of the spatial variability of precipitation. COMEPHORE relies on kriging for stratiform precipitation and radar  observations for convective precipitation.  While COMEPHORE uses isotropic covariance in

the kriging step,  anisotropy is introduced through the radar component. This suggests that  anisotropic variograms, particularly when informed by AROME fields, help better reproduce the spatial structure of radar-derived precipitation patterns.

[Figure]

**Figure 3.** Distribution of TWS scores for deterministic models (SPAZM) and stochastic models (rgISO, rgANISO, arISO, arANISO). Low values of TWS scores indicate better image gradients, using COMEPHORE fields as references. The distributions of TWS scores are plotted for low estimated anisotropy ratio ($0.75 \leq \eta \leq 1$, left), medium estimated anisotropy ratio ($0.5 \leq \eta \leq 0.75$, center), and strong estimated anisotropy ratio ($0 \leq \eta \leq 0.55$, right). The number of events is displayed per event class. For the stochastic models, the evaluation is conducted over the mean of 100 ensemble members of conditional simulations.

Fig. 4 presents the spectral slopes of the reference and predicted precipitation fields. SPAZM systematically underestimates the highest slopes, indicating an overestimation of fine-scale spatial variability. Compared to rgISO, rgANISO degrades the representation of spatial variability for weakly anisotropic events and provides only limited improvement for strongly anisotropic events. In contrast, arISO and arANISO improve the representation of spatial variability, particularly for strongly anisotropic events. Among them, arANISO better reproduces spatial variability, as the highest spectral slopes are no longer underestimated.

[Figure]

**Figure 4.** Scatter plots of the slopes of the radius averaged spectral density, using COMEPHORE data (reference) and the other gridded precipitation products models (SPAZM, rgISO, rgANISO, arISO, arANISO). The scatter plots of slopes are plotted for low estimated anisotropy ratio ($0.75 \leq \eta \leq 1$, top), medium estimated anisotropy ratio ($0.5 \leq \eta \leq 0.75$, center), and strong estimated anisotropy ratio ($0 \leq \eta \leq 0.55$, bottom). The dashed line represents the diagonal. Best model are highlighted per anisotropy class and metrics (ME, RMSE, best-fit line). For the stochastic models, the slopes are computed over the mean of 100 ensemble members of conditional simulations.

**4.3 Probabilistic performances and Uncertainty Quantification**

To assess the reliability of ensemble simulations, we apply the same cross-validation approach used for evaluating ensemble means (see Subsection 4.1). The Continuous Ranked Probability Score (CRPS) is computed to quantify probabilistic prediction accuracy, where lower values indicate better performance. Distribution of CRPS scores reveals similar performances for rgISO, rgANISO, and arISO but a notable improvement with arANISO (Fig.5). The error reduction is substantial for strong anisotropy events. This suggests that integrating anisotropy into the covariance structure using AROME enhances the ability of the ensemble to capture observed daily precipitation. Figure S1 in the Supplementary Materials demonstrates that this result holds across all weather regimes. Figure S2 further shows that the result also applies to lower precipitation intensity ranges

325  (1–20 mm and 20–50 mm). For this reason, we later quantify precipitation uncertainty using conditional simulations from anisotropic covariance, derived from AROME simulations.

[revised manuscript text omitted]

of the covariance (Paciorek and Schervish, 2006; Risser and Calder, 2017; Risser et al., 2019). For computational reasons, we decided not to include non-stationarity. Extending the method to larger and topographically complex domains would require non-stationary covariance. A practical option is to partition the region into climatologically homogeneous sub-regions, ideally preserving major watershed boundaries to maintain hydrological consistency. Alternatively, a solution is to incorporate non-stationary covariance structures, for example, through geographical coordinate deformation (Youngman, 2023) or locally stationary covariance models (Paciorek and Schervish, 2006; Risser and Calder, 2017), which would allow spatial dependence to evolve smoothly across the domain. These approaches would make the method suitable for operational applications over larger domains.

Despite those limitations, we are confident in the study's transposability to other regions of similar size with high-quality CP-RCM simulations. Furthermore, the scientific community should explore whether the same methodology can be applied using RCM simulations in regions where CP-RCM simulations are unavailable. The methodology  could also be extended to real-time, sub-daily interpolation. CP-RCM simulations are not continuously updated, so their replacement by hourly numerical weather forecasts (NWP) should be investigated. NWP assimilate past radar reflectivity and should therefore display a higher correlation with observations than CP-RCM simulations. As a result, NWP could allow us to extract precipitation intensity, spatial patterns, and spatial variability, while quantifying interpolation uncertainty through conditional simulations and the use of ensemble NWP forecasts. A natural follow-up would be to use NWP forecasts as both drift and covariance structures within a kriging-with-external-drift framework (Velasco-Forero et al., 2009; Schiemann et al., 2011). At the daily timescale, timesteps are typically considered independent, but this assumption no longer holds at the hourly scale. To address this, temporal dependence should be incorporated into the model, as done in Sideris et al. (2014); Frey and Frei (2025). A major limitation is the limited availability of sub-daily rain gauges. One potential solution to bypass this shortcoming is to disaggregate daily interpolated fields using radar data or 
[revised manuscript text omitted]

Youngman, B. D.: deform: An R Package for Nonstationary Spatial Gaussian Process Models by Deformations and Dimension Expansion, arXiv preprint arXiv:2311.05272, 2023.

650   Zandi, O., Zahraie, B., Nasseri, M., and Behrangi, A.: Stacking machine learning models versus a locally weighted linear model to generate high-resolution monthly precipitation over a topographically complex area, Atmospheric Research, 272, 106 159, https://doi.org/10.1016/j.atmosres.2022.106159, 2022.